# Graph-Link: Bridging the Semantic-Structural Gap in Text-to-SQL via Constrained Subgraph Induction

Jianwei Zhong [1]   Yuxi Yang [1]   Quanxin Liu [1]   Ruida Xu [1]   Ming Chi [1]   Yijun Mo [1]

## Abstract

Schema Linking serves as the foundational perception layer in Text-to-SQL, tasked with grounding natural language queries into relevant schema elements. However, existing retrieval-based approaches suffer from a critical *structural blindness*: by prioritizing elements with high textual similarity, they inadvertently prune semantically-thin but topologically-critical bridge tables, thereby severing relational pathways necessary for multi-hop joins. To bridge this gap, we propose Graph-Link, a novel framework that reformulates schema linking from an independent retrieval task into a constrained subgraph induction problem. We argue that generating executable SQL necessitates a connected subgraph that satisfies both semantic relevance and structural constraints. Accordingly, Graph-Link employs a hierarchical schema graph to model the search space across multiple granularities, and then applies a Steiner-tree-based optimization for subgraph induction that guarantees the topological connectivity while maximizing the signal-to-noise ratio for downstream LLMs. Extensive experiments on BIRD and Spider 2.0 demonstrate that Graph-Link achieves state-of-the-art schema linking performance, improving recall and hit rates by up to 7.0% over competitive baselines, and boosts downstream SQL generation accuracy on complex queries by 13.8%.

## 1. Introduction

Text-to-SQL stands as a pivotal interface in the democratization of data analytics, enabling non-expert users to query massive relational databases using natural language without mastering complex query languages (Li et al., 2023b). With the advances of large language models (LLMs), the paradigm of Text-to-SQL has shifted from specialized supervised models (Hui et al., 2023; Li et al., 2025a) to LLM-based agents utilizing In-Context Learning (ICL).These agentic approaches leverage the reasoning capabilities of LLMs to generate SQL by treating the database schema as part of the prompt context (Gao et al., 2023; Dong et al., 2023). Although proficient on academic benchmarks, their efficacy diminishes in real-world deployment scenarios (Lei et al., 2024). Production databases often contain hundreds of tables with thousands of columns, creating a context availability dilemma: the full schema far exceeds the context window of standard LLMs, while the inclusion of irrelevant metadata exacerbates the risk of hallucinations and reasoning failures (Liu et al., 2024).

Consequently, Schema Linking has emerged as the foundational perception layer, tasked with distilling the massive database into a subset of pertinent tables and columns (Lee et al., 2025). Current state-of-the-art approaches typically address this bottleneck by employing multi-step retrieval or sampling pipelines (Li et al., 2023b). These methods often utilize dense retrieval or LLM-based selection to filter out irrelevant schema items, aiming to construct a condensed schema that fits within the context window while retaining necessary information (Lin et al., 2023; Li et al., 2025b).

However, current paradigms face a critical dichotomy between semantic relevance and structural integrity (Safdarian et al., 2025). As analyzed in Table 1, we identify three primary failure modes: (1) **Structural Blindness**, where retrieval-based methods sever foreign key pathways by discarding semantically latent bridge tables (Cao et al., 2021); (2) **Reasoning Unfaithfulness** in agentic workflows, characterized by hallucinations and semantic drift; and (3) **Local Optimization**, where step-wise filtering fails to guarantee the global graph connectivity essential for multi-hop queries.

Recent empirical evidence quantifies a widening perception gap in large-scale deployments, where schema linking deficits drive over 60% of execution errors (Wang et al., 2025c). These errors manifest primarily as structural disintegration: aggressive pruning severs foreign key path-

---

[1]Huazhong University of Science and Technology, Wuhan, China. Correspondence to: Yijun Mo <moyj@hust.edu.cn>, Yuxi Yang <yx_yang@hust.edu.cn>.

*Table 1.* Analysis of primary directions in Text-to-SQL Schema Linking.

| Reference | Contribution | Common Weak Points | | |
|---|---|---|---|---|
| | | Structural Blindness | Reasoning Unfaithfulness | Local Optimization |
| Hybrid (BM25+Dense) | Dual-Stream Retrieval | ✓ (Semantic Gap) | × | ✓ (Independent Ranking) |
| DIN-SQL (Pourreza & Rafiei, 2023) | Chain-of-Thought | ✓ (Linear Pruning) | ✓ (Invalid CoT Steps) | ✓ (Step-wise Filter) |
| CHESS (Talaei et al., 2024) | Hierarchical Pruning | ✓ (Agent Pruning) | ✓ (Contextual Hallucination) | × |
| CHASE-SQL (Pourreza et al., 2024) | Multi-Path Reasoning | ✓ (Linear Pruning) | ✓ (Generative Bias) | ✓ (Step-wise Filter) |
| MAC-SQL (Wang et al., 2025a) | Multi-Agent Collab. | ✓ (Agent Isolation) | ✓ (Semantic Drift) | × |
| LitE-SQL (Piao et al., 2025) | Vector-based Retrieval | ✓ (Semantic Gap) | × | ✓ (Independent Ranking) |
| MCS-SQL (Lee et al., 2025) | Multiple Choice Sel. | × | ✓ (Stochastic Noise) | × |
| JOLT-SQL (Song et al., 2025) | Joint Loss Tuning | ✓ (Attention Span Limit) | × | × |
| SGU-SQL (Zhang et al., 2024) | Graph Attention | ✓ (Learned Alignment) | × | ✓ (Contrastive Local Opt.) |
| SchemaGraphSQL (Safdarian et al., 2025) | Pairwise Path Enum. | × | ✓ (LLM Hallucination) | ✓ (Pairwise Heuristic) |
| **Ours (Graph-Link)** | **Steiner Tree Reasoning** | × | × | × |

ways, discarding intermediate tables essential for multi-hop `JOIN`s. This creates a fatal asymmetry where LLMs, despite being robust to schema noise (false positives), remain brittle to omission errors: as noted by Maamari et al. (2024), missing a single required column guarantees execution failure. Consequently, the prevailing "filter-then-generate" paradigm faces a fundamental paradox: optimizing for local semantic relevance comes at the cost of the global topological connectivity required for executable SQL (Yao et al., 2023).

To resolve these challenges simultaneously, we seek to bridge the semantic-structural gap by reformulating schema linking not as a retrieval task, but as a path-reasoning problem on a graph. Drawing inspiration from structural homology, we observe that a valid SQL query essentially corresponds to a connected subgraph that satisfies both semantic relevance (matching user intent) and topological constraints (adhering to foreign keys) (Ma et al., 2024; Luo et al., 2023). This perspective shifts the objective from local keyword matching to finding an optimal communication path between data entities (Pan et al., 2024).

Based on this observation, we propose Graph-Link, a framework that synergizes semantic retrieval with topological pathfinding to construct a minimal connected subgraph that faithfully represents the user's intent. Specifically, we first explicitly model the database schema as a semantic graph, where tables and columns serve as nodes connected by foreign key dependencies. Upon this topological backbone, Graph-Link utilizes hierarchical community summaries to maintain global awareness of the database structure, effectively preventing the tunnel vision typical of local retrieval methods (Edge et al., 2024). Furthermore, we formulate context selection as a Steiner-tree-based optimization problem (Yao et al., 2023). This mathematical formulation guarantees structural connectivity, allowing the model to recover implicit bridge tables that purely semantic matching would miss. By integrating community detection with topological pathfinding, Graph-Link effectively grounds natural language into the relational structure of the database. We conduct comprehensive evaluations on the cross-domain

BIRD and Spider 2.0 benchmarks, comparing Graph-Link against both hybrid retrieval methods and recent state-of-the-art agentic frameworks.

Our major contributions are summarized as follows:

- We identify the structural limitations in existing retrieval-based schema linking methods and redefine the task as a graph path-reasoning problem, emphasizing the necessity of topological connectivity for valid SQL generation.

- We propose Graph-Link, a novel framework that combines hierarchical schema graph for global context awareness with a Steiner-tree optimization algorithm to ensure the retrieval of a connected, semantically relevant subgraph.

- We conduct comprehensive evaluations on BIRD and Spider 2.0, demonstrating that Graph-Link establishes a new state-of-the-art with a 90.8% schema linking hit rate. Crucially, our analysis reveals that enforcing topological connectivity effectively mitigates structural blindness, yielding a 13.8% improvement in downstream execution accuracy on complex multi-hop queries compared to leading agentic baselines.

## 2. Related Work

### 2.1. Retrieval-Based Schema Linking

The prevailing paradigm treats schema linking as a relevant item retrieval task to manage large-scale databases. Early approaches utilized sparse retrieval techniques, such as BM25 and n-gram matching, to align natural language tokens with schema names (Robertson & Zaragoza, 2009; Katsogiannis-Meimarakis & Koutrika, 2023). With the advent of Retrieval Augmented Generation(RAG) (Lewis et al., 2020), dense retrieval methods (e.g., DPR, Contriever) have been widely adopted to map queries and schema elements into a shared embedding space, ranking tables based on semantic similarity (Li et al., 2023b;a; 2025b). These methods

efficiently filter out clearly irrelevant tables. However, by prioritizing textual similarity, they risk overlooking "bridge tables"—intermediate relations that lack semantic overlap with the user query but are topologically essential for multi-hop joins (Wang et al., 2025b).

## 2.2. Agentic Frameworks

The emergence of Large Language Models (LLMs) has shifted the focus towards agent-based reasoning. Frameworks like DIN-SQL (Pourreza & Rafiei, 2023) and MAC-SQL (Wang et al., 2025a) employ In-Context Learning (ICL) and Chain-of-Thought (CoT) (Wei et al., 2022) prompting to simulate the reasoning process of human experts.

These approaches typically serialize a subset of the schema into the prompt and leverage the LLM's self-attention to infer table relationships and select relevant columns iteratively. While these generative methods achieve high accuracy on complex benchmarks, they often incur high latency and token costs due to multi-turn prompting. Furthermore, injecting massive schema contexts into probabilistic LLMs can trigger the "lost-in-the-middle" phenomenon (Long et al., 2025), where reasoning accuracy degrades as context grows, leading to hallucinations of plausible but non-existent schema links that violate database constraints (Lee et al., 2025; Liu et al., 2024).

## 2.3. Graph-Based Structural Reasoning

Incorporating graph structures has long been a core direction in Text-to-SQL. Prior to the LLM era, Graph Neural Networks (GNNs), such as RAT-SQL (Wang et al., 2020) and LGESQL (Cao et al., 2021), were standardly used to encode the database schema graph and link it to the question utterance. More recently, SGU-SQL (Zhang et al., 2024) employs structure-guided relational graph attention for schema linking, while SchemaGraphSQL (Safdarian et al., 2025) utilizes pathfinding algorithms on schema graphs to identify join paths. Despite their graph-based nature, these approaches either rely on trained parametric encoders or resort to pairwise shortest-path heuristics, lacking a globally optimal connectivity guarantee.

More recently, Graph Retrieval-Augmented Generation (GraphRAG) (Edge et al., 2024) has applied graph traversals and Steiner Tree optimizations to open-domain question answering (Bhalotia et al., 2002). Our work sits at the intersection of these fields: distinct from training heavy GNN encoders, we adapt the optimization logic of Steiner Trees as a lightweight, inference-time mechanism. This allows us to strictly enforce the topological connectivity required for SQL execution while leveraging the semantic power of modern LLMs.

## 3. Preliminaries

### 3.1. Task Definition

Let $\mathcal{D} = (\mathcal{T}, \mathcal{C}, \mathcal{R})$ denote a relational database schema, comprising a set of tables $\mathcal{T} = \{t_1, \ldots, t_N\}$, a set of columns $\mathcal{C} = \{c_1, \ldots, c_M\}$, and a set of relational constraints $\mathcal{R}$ which includes primary-foreign key pairs. Given a natural language query $Q = \{x_1, \ldots, x_L\}$ representing the user's intent, the objective of Text-to-SQL is to generate an executable SQL query $Y$ that is semantically equivalent to $Q$ with respect to $\mathcal{D}$. Within this framework, Schema Linking functions as a prerequisite perception task. It seeks to identify a relevant sub-schema $\mathcal{D}^* \subseteq \mathcal{D}$ (where $|\mathcal{D}^*| \ll |\mathcal{D}|$) that covers the intent of $Q$ while adhering to the relational constraints required for valid SQL execution (Long et al., 2025).

### 3.2. Schema Graph Definition

To bridge the modality gap between the linear structure of natural language and the relational topology of databases, we unify the schema representation into a **Table-Centric Heterogeneous Graph**, denoted as $\mathcal{G} = (\mathcal{V}, \mathcal{E}, \mathbf{W})$. The node set $\mathcal{V} = \mathcal{T} \cup \mathcal{C}$ is functionally partitioned: table nodes $\mathcal{T}$ serve as the *topological backbone*, acting as the primary junctions for structural navigation and pathfinding, while column nodes $\mathcal{C}$ function as *semantic leaves* attached to their respective tables, responsible solely for grounding user intent via dense retrieval without participating in graph traversal. Accordingly, the edge set $\mathcal{E}$ comprises undirected structural edges connecting table pairs based on foreign key constraints to model valid `JOIN` pathways, and affiliation edges linking columns to their parent tables. This structure allows us to decouple the reasoning process, where semantic relevance is computed at the column level and aggregated, while connectivity optimization is performed exclusively on the table-weighted topology via the resistance matrix $\mathbf{W}$.

### 3.3. Problem Reformulation

Existing approaches predominantly model schema linking as a *Pointwise Ranking* problem, selecting elements that maximize independent semantic probabilities under cardinality or confidence constraints, i.e., $\mathcal{D}^* = \arg\max_{\mathcal{D}' \subseteq \mathcal{D}} \sum_{v \in \mathcal{D}'} \log P(v \mid q)$ such that $|\mathcal{D}'| \leq k$ or $\max_{v \in \mathcal{D}'} P(v \mid q) > \tau$. While computationally efficient, this independence assumption leads to a fundamental topological defect:

**Definition 3.1** (**Structural Blindness**)**.** Let $\pi(u, v)$ be the unique valid `JOIN` path between two relevant nodes $u, v$. A model exhibits *Structural Blindness* if it discards a bridge node $k \in \pi(u, v)$ solely because $k$ lacks lexical overlap with $q$ (i.e., $P(k|q) < \tau$), thereby rendering $\pi(u, v)$ disconnected and the resulting SQL inexecutable.

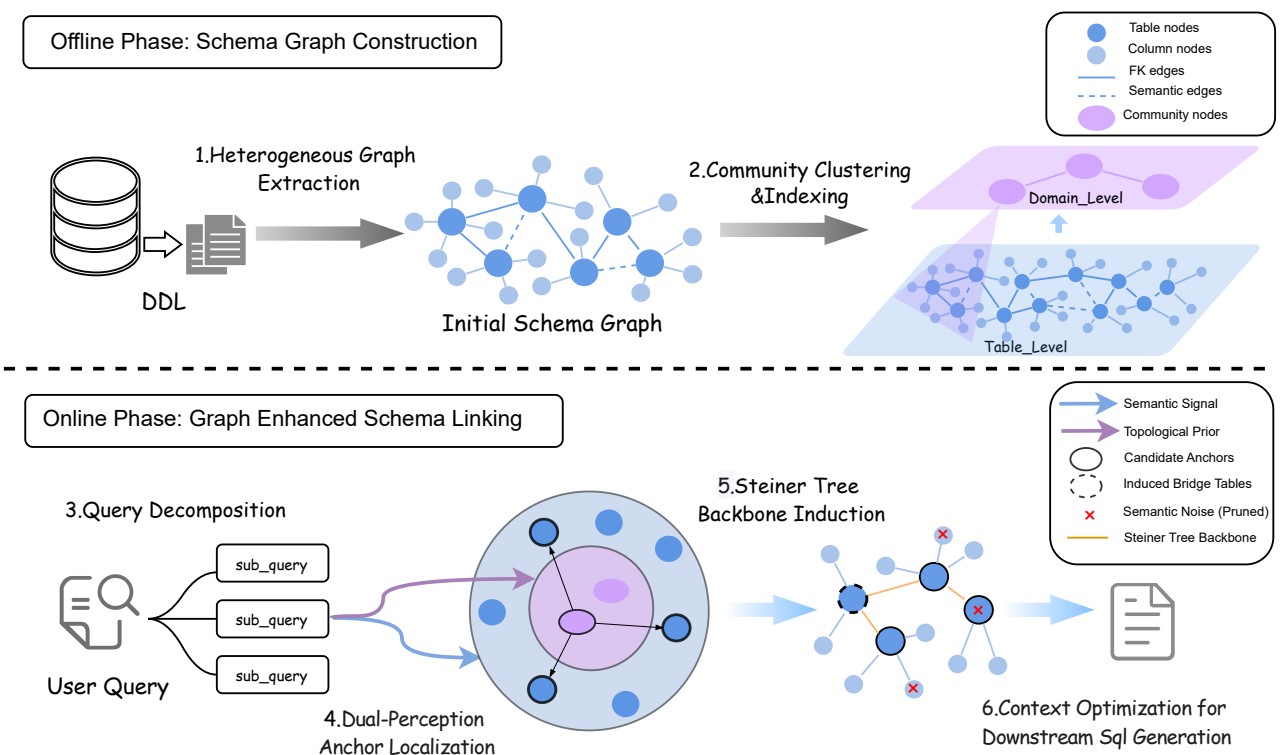

*Figure 1.* Graph-Link workflow: (1) Offline: Construct a table-centric heterogeneous graph where nodes represent tables/columns and edges encode foreign key constraints. Partition into communities via Leiden algorithm for hierarchical indexing. (2) Online: (a) Decompose query into sub-queries guided by community summaries; (b) Identify anchor tables via dual scoring (semantic similarity + topological centrality); (c) Induce Steiner tree backbone to recover bridge tables (blue nodes); (d) Prune irrelevant columns while preserving primary/foreign keys. The final output is a minimal connected subgraph sufficient for SQL generation.

To resolve Structural Blindness, we reformulate schema linking from stochastic retrieval into a deterministic **Constrained Subgraph Induction** problem. Instead of maximizing node relevance, we minimize the joint potential of a connected subgraph. Mathematically, this maps to the Node-Weighted Steiner Tree (NWST) problem:

$$\mathcal{G}^* = \underset{\mathcal{G}' \subseteq \mathcal{G}}{\arg\min} \left( \sum_{v \in \mathcal{V}(\mathcal{G}')} \psi_{\text{sem}}(v, q) + \lambda \sum_{e \in \mathcal{E}(\mathcal{G}')} w_{\text{res}}(e) \right) \tag{1}$$

Here, $\psi_{\text{sem}}(v, q) \propto -\log P(v|q)$ represents the semantic cost, and $w_{\text{res}}(e)$ denotes edge resistance. This objective forces the inclusion of high-cost (low-relevance) bridge nodes if they offer the path of least resistance between anchors, guaranteeing a topologically valid $\mathcal{D}^*$.

Equation (1) defines a conceptual objective; directly optimizing the NWST is NP-hard. Graph-Link therefore approximates this ideal through a three-stage pipeline: (i) anchor selection to ground semantically relevant tables, (ii) Steiner tree backbone induction to recover bridge tables and enforce connectivity, and (iii) LLM-critic refinement to prune irrelevant columns. Algorithm 1 outlines this approximation.

## 4. Methodology

We propose **Graph-Link**, a framework that reformulates schema linking from a stochastic retrieval task into a deterministic *Constrained Subgraph Induction* problem. At a high level, Graph-Link is grounded in three core observations: (1) executable SQL queries inherently require a *connected* schema subgraph rather than a set of independently relevant tables; (2) local semantic retrieval alone is insufficient for preserving multi-hop relational pathways in large schemas; and (3) semantically silent but structurally indispensable bridge tables must be recovered through explicit topological reasoning. These observations directly motivate our design choices: hierarchical schema abstraction for global awareness, dual-perception anchor localization for robust grounding, and Steiner-tree-based optimization to guarantee structural connectivity.

As illustrated in Figure 1, Graph-Link adopts a decoupled dual-phase architecture. The *offline phase* constructs a hierarchically indexed schema graph that encodes global structural priors and ensures scalability to real-world databases. The *online phase* performs topology-constrained subgraph induction over a focused view of the schema, explicitly ad-

**Algorithm 1** Graph-Link: Topology-Constrained Subgraph Induction

**Require:** Database schema $\mathcal{D}$, Query $q$, thresholds $\tau, \delta_{prune}$
**Ensure:** Connected sub-schema $\mathcal{D}^*$
 1: {**Phase I: Offline Hierarchical Graph Construction**}
 2: Initialize $\mathcal{G}$ based on definitions in Sec. 3.2
 3: **for all** pairs $(t_i, t_j) \in \mathcal{T} \times \mathcal{T}$ **do**
 4:     Compute resistance $w_{ij}$ via Eq. 3
 5:     **if** $(t_i, t_j) \in \mathcal{R}$ OR $w_{ij} < 1 - \tau$ **then**
 6:         Update edge $e_{ij} \in \mathcal{E}$ with weight $w_{ij}$
 7:     **end if**
 8: **end for**
 9: $\mathbb{C} \leftarrow \text{LeidenPartition}(\mathcal{G})$ {Maximize modularity $Q$}
10: {**Phase II: Online Subgraph Induction**}
11: $\mathcal{Q}_{sub} \leftarrow \text{LLM-Decompose}(q, \text{Summaries}(\mathbb{C}))$
12: $\mathcal{G}_{view} \leftarrow \text{Route}(\mathcal{Q}_{sub}, \mathbb{C}) \cup \mathcal{E}_{inter}$
13: $\mathcal{A} \leftarrow \{t \in \mathcal{V}(\mathcal{G}_{view}) \mid \mathcal{S}(t) > \text{Threshold}\}$
14: {**Steiner Tree Backbone Induction**}
15: $G_D \leftarrow \text{MetricClosure}(\mathcal{A}, \mathcal{G}_{view})$
16: $T_{MST} \leftarrow \text{Prim-MST}(G_D)$
17: $\mathcal{T}^* \leftarrow \text{MapToPhysicalPaths}(T_{MST}, \mathcal{G}_{view})$
18: {**Refined Context Optimization**}
19: $\mathcal{D}^* \leftarrow \text{LLM-Refine}(\mathcal{T}^*, \text{PK} \cup \text{FK}, \delta_{prune})$
20: **return** $\mathcal{D}^*$

dressing the structural blindness inherent in independent retrieval. The complete procedure is summarized in Algorithm 1.

### 4.1. Offline Phase: Hierarchical Schema Graph Construction

**Design Rationale.** A naive Steiner-tree search over a flat schema graph is computationally infeasible for databases containing hundreds of tables. More critically, without a global structural prior, local semantic retrieval tends to over-explore dense but irrelevant regions of the schema graph, leading to myopic pruning decisions. We therefore introduce a hierarchical abstraction that compresses the global topology into semantically coherent communities. This hierarchy does not merely accelerate inference: it establishes a structural prior that constrains online pathfinding to a tractable and semantically aligned search space.

**Heterogeneous Graph Extraction.** We instantiate the abstract table-centric graph $\mathcal{G} = (\mathcal{V}, \mathcal{E}, \mathbf{W})$ defined in Section 3.2. For each table node $t_i \in \mathcal{V}$, we synthesize a semantic representation $\mathbf{g}_{t_i}$ by encoding the concatenation of the table name and description using a pre-trained dense encoder (e.g., `text-embedding-v4`). Column embeddings $\{\mathbf{h}_c\}$ are derived analogously from full column names, descriptions, and sample values.

To quantify the resistance of traversing between tables, we define a *Dual-Scale Similarity* between table pairs:

$$S_{ij} = \alpha\,\sigma(\mathbf{g}_{t_i}, \mathbf{g}_{t_j}) + (1-\alpha) \max_{c \in \text{Col}(t_i),\, c' \in \text{Col}(t_j)} \sigma(\mathbf{h}_c, \mathbf{h}_{c'}), \tag{2}$$

where $\sigma(\cdot, \cdot)$ denotes cosine similarity and $\alpha \in [0, 1]$ balances holistic table semantics against localized column-level correspondences. This design is motivated by the observation that join-relevant tables often share weak global semantics while exhibiting strong localized overlaps through identifier or foreign-key attributes.

We convert similarity into traversal resistance via:

$$w_{ij} = \begin{cases} \epsilon & \text{if } (t_i, t_j) \in \mathcal{E}_{\text{FK}}, \\ 1 - S_{ij} & \text{if } (t_i, t_j) \notin \mathcal{E}_{\text{FK}} \wedge S_{ij} > \tau, \\ \infty & \text{otherwise}, \end{cases} \tag{3}$$

where $\mathcal{E}_{\text{FK}}$ denotes primary–foreign key relations. By setting $\epsilon \to 0$, explicit relational constraints become effectively "superconductive", ensuring that valid join paths are always preferred during optimization unless overridden by strong semantic evidence.

**Community-Based Hierarchical Indexing.** To further control the search space, we partition $\mathcal{G}$ into semantically coherent communities $\mathbb{C} = \{C_1, \ldots, C_K\}$. We transform resistance weights into affinities via $A_{ij} = \exp(-w_{ij})$ and apply the Leiden algorithm (Traag et al., 2019) to maximize modularity. Each community is summarized by (1) a *Geometric Centroid* $\mathbf{s}_k = \frac{1}{|C_k|} \sum_{t \in C_k} \mathbf{g}_t$ for vector-based routing, and (2) a *Semantic Synopsis* generated by an LLM to capture the functional domain of the cluster. The prompt used for community summarization is provided in Appendix. This dual-modal index enables efficient coarse-to-fine pruning during online inference.

### 4.2. Online Phase: Steiner-Tree-Based Subgraph Induction

The online phase addresses the *independence fallacy* of pointwise retrieval by enforcing global connectivity constraints during schema selection.

**Schema-Guided Query Decomposition and Routing.** Given a natural language query $q$, we employ an LLM agent (refer to Appendix C for detailed prompt templates) to decompose the input query into a set of atomic sub-queries $\mathcal{Q}_{sub} = \{q_1, \ldots, q_m\}$, guided by the community summaries. These sub-queries act as routing probes to activate relevant schema regions:

$$\mathbb{C}_{\text{active}} = \left\{ C_k \in \mathbb{C} \,\middle|\, \max_{q_i \in \mathcal{Q}_{sub}} \sigma(\mathbf{e}_{q_i}, \mathbf{s}_k) > \tau_{\text{route}} \right\}, \tag{4}$$

We then construct a focused candidate view $\mathcal{G}_{view}$ by retrieving all tables within $\mathbb{C}_{active}$ along with their cross-community edges, filtering out irrelevant domains prior to fine-grained inference.

**Dual-Perception Anchor Localization.** Within $\mathcal{G}_{view}$, we identify *anchor tables* that explicitly ground the user intent. Pure semantic matching tends to over-select peripheral tables, while purely structural heuristics favor hubs lacking semantic relevance. To balance these failure modes, we compute a dual-perception relevance score:

$$\mathcal{S}(t) = \underbrace{\max_{k=1}^{m} \max_{c \in \text{Col}(t)} \sigma(\mathbf{e}_{q_k}, \mathbf{h}_c)}_{\text{Semantic Signal}} \cdot \underbrace{\left(1 + \lambda \log \deg_{\mathcal{G}}(t)\right)}_{\text{Topological Prior}},$$

(5)

where $\sigma(\cdot)$ denotes cosine similarity and $\deg_{\mathcal{G}}(t)$ represents the node's global degree. The max-pooling operation ensures that a table is flagged as relevant if it contains even a *single* column matching a sub-query, preventing dilution from irrelevant attributes. The final anchor set $\mathcal{A}$ is formed by selecting nodes where $\mathcal{S}(t)$ exceeds a dynamic threshold.

**Steiner-Tree Backbone Induction.** The central challenge of schema linking lies in recovering structurally necessary but semantically silent bridge tables. The Steiner Tree formulation explicitly optimizes for the minimal connected structure spanning all anchors, making it uniquely suited for this task.

Accordingly, we model this as a Node-Weighted Steiner Tree (NWST) problem. Our objective is to induce a connected subgraph $\mathcal{T}^* = (\mathcal{V}^*, \mathcal{E}^*)$ that connects all anchors in $\mathcal{A}$ with minimum total resistance:

$$\min_{\mathcal{T}^* \subseteq \mathcal{G}_{view}} \sum_{e \in \mathcal{E}^*} w(e) \quad \text{s.t.} \quad \mathcal{A} \subseteq \mathcal{V}^*. \tag{6}$$

Foreign-key edges receive negligible resistance, ensuring that valid relational paths dominate the solution.

Since NWST is NP-hard, exact inference is intractable for online deployment. However, the problem becomes practical in our setting due to two properties: (i) aggressive hierarchical pruning restricts optimization to a compact $\mathcal{G}_{view}$, and (ii) the anchor set $|\mathcal{A}|$ is typically small. We therefore adopt a standard **2-approximation via metric closure**. Specifically, we construct a complete graph $G_D = (\mathcal{A}, \mathcal{E}_D)$ over anchors, where each virtual edge $(u, v)$ is weighted by the shortest-path distance between $u$ and $v$ in $\mathcal{G}_{view}$, computed using Dijkstra's algorithm. We then compute a Minimum Spanning Tree (MST) on $G_D$ to obtain the optimal connectivity skeleton among anchors.

To materialize the executable schema backbone, each virtual MST edge is mapped back to its corresponding physical path in $\mathcal{G}_{view}$, and the union of these paths yields

$\mathcal{T}^*$. This reverse projection step explicitly reinstates all intermediate tables on the shortest paths, thereby recovering the missing bridge tables that are structurally required but semantically ungrounded. The overall complexity is dominated by anchor-restricted shortest-path computation, $O(|\mathcal{A}| \cdot |\mathcal{E}_{view}| \log |\mathcal{V}_{view}|)$, which is negligible in practice and consistently lower than downstream LLM inference latency.

**LLM-Critic Refined Context Optimization.** Finally, we project the table-level backbone into a column-level executable schema. An LLM critic prunes irrelevant attributes based on semantic scores while strictly preserving all primary and foreign keys induced by the Steiner backbone. The final schema $\mathcal{D}^*$ is:

$$\mathcal{D}^* = \bigcup_{t \in \mathcal{V}(\mathcal{T}^*)} \Big( \{c \in t \mid c \in \text{PK} \cup \text{FK}\} \\ \cup \{c \in t \mid \mathcal{S}_{col}(c) > \delta_{prune}\} \Big) \tag{7}$$

This yields a minimal, structurally complete schema context that maximizes the signal-to-noise ratio for downstream SQL generation.

## 5. Experiments

### 5.1. Experimental Setup

**Datasets.** We evaluate on two cross-domain benchmarks: BIRD (Dev) (Li et al., 2023b) and Spider 2.0-Lite (Lei et al., 2024), covering both large-scale and standard evaluation settings. Detailed statistics and schema characteristics are provided in Appendix A.1.

**Baselines & Models.** We compare Graph-Link with general retrieval methods (Zero-shot, Hybrid BM25+Vector) and SOTA agentic frameworks (DIN-SQL, MAC-SQL, MCS-SQL, RSL-SQL). All methods are evaluated using a unified configuration: `text-embedding-v4` (Zhang et al., 2025) serves as the backbone for dense representation, while Qwen3-max (Yang et al., 2025) and GLM-4.7 (GLM et al., 2024) function as the inference engines for reasoning and generation. Complete implementation details and hyperparameter settings are listed in Appendix A.

**Metrics.** We report Precision, Recall, F1, and **Hit Rate** (percentage of samples with full gold schema retrieval). We prioritize **F2-score** ($\beta = 2$) to strictly penalize false negatives, as missing schema elements renders generation impossible. End-to-end performance is measured by **Execution Accuracy (EX)** and token usage.

### 5.2. Schema Linking Performance

Table 2 compares schema linking performance on the BIRD (Dev) and Spider 2.0-Lite benchmarks. Graph-Link

*Table 2.* **Main Results: Schema Linking Performance.** We compare the effectiveness of identifying the correct schema elements on BIRD (Development Set) and Spider 2.0-Lite. Best results are **bold**, second best are underlined.

| Method | Reference Model | BIRD-Dev (2025) (Li et al., 2023b) | | | | | SPIDER2.0-Lite (Lei et al., 2024) | | | | |
|---|---|---|---|---|---|---|---|---|---|---|---|
| | | Prec. | Rec.(↑) | Hit(↑) | F1 | F2(↑) | Prec. | Rec. | Hit | F1 | F2 |
| *General Retrieval Methods* | | | | | | | | | | | |
| Zero-shot Prompting | Qwen3-max | 86.6 | 76.8 | 46.7 | 81.4 | 78.6 | 79.6 | 71.5 | 40.0 | 75.3 | 73.0 |
| | GLM-4.7 | 84.2 | 77.3 | 47.9 | 80.6 | 78.6 | 77.6 | 72.1 | 42.2 | 74.4 | 72.6 |
| Hybrid (BM25 + Vector) | text-embed-4 | 17.2 | 61.9 | 33.5 | 26.9 | 40.7 | 16.3 | 62.1 | 26.3 | 25.8 | 39.8 |
| *Agentic & SOTA Frameworks* | | | | | | | | | | | |
| DIN-SQL (Pourreza & Rafiei, 2023) | Qwen3-max | 86.3 | 79.0 | 57.7 | 82.5 | 80.3 | **88.2** | 77.8 | 53.4 | **82.7** | 79.7 |
| | GLM-4.7 | 84.3 | 80.2 | 58.6 | 82.2 | 81.0 | 80.6 | 83.8 | 59.4 | 82.2 | 83.1 |
| MAC-SQL (Wang et al., 2025a) | Qwen3-max | **87.4** | 81.1 | 61.1 | 84.1 | 82.3 | 82.3 | 78.6 | 59.3 | 80.4 | 79.3 |
| | GLM-4.7 | 86.5 | 83.4 | 63.2 | **84.9** | 84.0 | 80.6 | 80.2 | 61.2 | 80.4 | 80.3 |
| MCS-SQL (Lee et al., 2025) | Qwen3-max | 54.9 | 91.4 | 81.3 | 68.6 | 80.7 | 63.6 | 83.1 | 69.1 | 72.3 | 78.8 |
| | GLM-4.7 | 53.6 | 90.5 | 80.6 | 67.3 | 79.5 | 59.8 | 85.3 | 74.3 | 69.6 | 77.1 |
| RSL-SQL (Cao et al., 2024) | Qwen3-max | 50.6 | 89.3 | 77.6 | 64.6 | 77.5 | 40.9 | 89.6 | 70.3 | 56.2 | 72.4 |
| | GLM-4.7 | 48.1 | 86.7 | 78.3 | 58.5 | 72.7 | 34.9 | **91.8** | 75.2 | 50.6 | 69.2 |
| *Ours* | | | | | | | | | | | |
| **Graph-Link** | **Qwen3-max** | 70.4 | **95.9** | **90.8** | 81.2 | **89.4** | 69.7 | 91.3 | **80.3** | 79.1 | 86.0 |
| | **GLM-4.7** | 75.5 | 92.6 | 88.5 | 83.2 | 88.6 | 72.7 | 90.4 | 76.7 | 80.6 | **86.2** |

achieves the best results across all evaluated metrics. On BIRD, Graph-Link with Qwen3-max reaches a Hit Rate of **90.8%**, improving over the strongest baseline, MCS-SQL, by 9.5 percentage points. With GLM-4.7, Graph-Link attains a Hit Rate of 88.5%, compared to 78.3% for RSL-SQL. Since the Hit Rate provides an upper bound on valid SQL generation, these gains indicate a substantial improvement in schema coverage. Graph-Link also obtains the highest Recall (**95.9%**) and F2-score (**89.4%**), significantly outperforming hybrid retrieval and agentic baselines.

Figure 2 shows the Precision–Recall trade-off. Agentic methods such as MAC-SQL exhibit higher Precision but lower Recall (below 85%), suggesting aggressive pruning of structurally required columns. Graph-Link favors topological completeness, which reduces missing-bridge errors and leads to higher F2 scores. Similar trends are observed with both Qwen and GLM backbones, indicating that the proposed formulation is largely model-agnostic.

### 5.3. Robustness Analysis

To verify the hypothesis that graph modeling resolves structural blindness, we analyze performance breakdowns by query complexity (defined by the number of tables in the ground truth SQL) in Table 3.

**Resistance to the Complexity Cliff.** Baseline methods exhibit significant performance degradation as query complexity increases. Standard Hybrid retrieval suffers a catastrophic drop in Hit Rate, falling from 42.3% on "Simple"

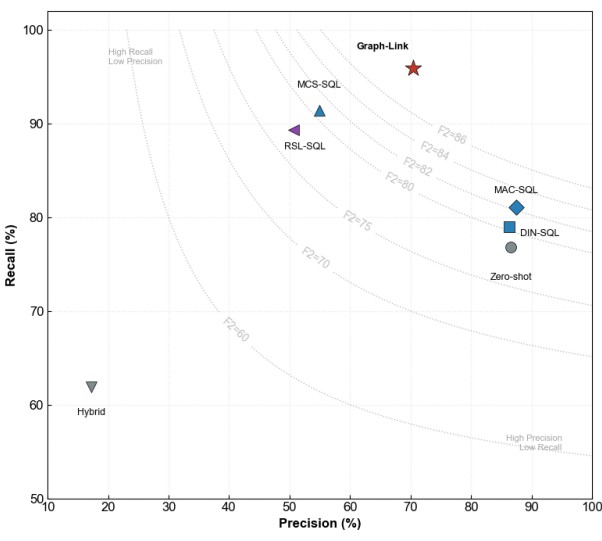

*Figure 2.* **Precision-Recall Trade-off on BIRD Dev.**

(1 table) to just 10.6% on "Challenge" (> 3 tables) queries. Similarly, zero-shot prompting drops to 27.7%. This confirms that semantic similarity alone is insufficient for multi-hop reasoning, as intermediate tables in complex joins often lack textual overlap with the query.

**Topology-Driven Robustness.** Conversely, as shown in Figure 3, Graph-Link demonstrates exceptional stability across complexity clusters. On "Challenge" queries, it maintains a **Hit Rate of 83.1%** and an **F2 Score of 83.8%**. This rep-

*Table 3.* Performance Breakdown by Query Complexity (BIRD Dev).

| Complexity | Method | Recall (↑) | Hit | F2 Score (↑) |
|---|---|---|---|---|
| **Simple** (1 table) | Zero-shot Prompting | 80.8 | 52.9 | 80.8 |
| | Hybrid (BM25 + Vector) | 70.2 | 42.3 | 41.3 |
| | DIN-SQL | 83.5 | 62.3 | 82.9 |
| | MAC-SQL | 84.8 | 65.6 | 84.9 |
| | MCS-SQL | 95.2 | 84.9 | 82.4 |
| | RSL-SQL | 93.2 | 80.5 | 78.4 |
| | **Graph-Link** | **96.6** | **90.2** | **90.8** |
| **Moderate** (2–3 tables) | Zero-shot Prompting | 75.2 | 40.4 | 78.5 |
| | Hybrid (BM25 + Vector) | 57.0 | 23.5 | 39.5 |
| | DIN-SQL | 74.3 | 56.7 | 76.9 |
| | MAC-SQL | 78.2 | 56.8 | 78.9 |
| | MCS-SQL | 87.4 | 78.3 | 78.5 |
| | RSL-SQL | 85.1 | 76.6 | 77.2 |
| | **Graph-Link** | **95.2** | **86.8** | **89.6** |
| **Challenge** (> 3 tables) | Zero-shot Prompting | 56.6 | 27.7 | 65.7 |
| | Hybrid (BM25 + Vector) | 25.7 | 10.6 | 40.5 |
| | DIN-SQL | 65.9 | 32.2 | 74.8 |
| | MAC-SQL | 67.4 | 46.6 | 76.5 |
| | MCS-SQL | 80.2 | 68.5 | 77.4 |
| | RSL-SQL | 78.6 | 62.6 | 72.8 |
| | **Graph-Link** | **93.7** | **83.1** | **83.8** |

resents a massive improvement over the closest competitor (MCS-SQL: 68.5% Hit). The gap is most pronounced in recalling multi-table relations: when linear retrieval cannot bridge semantic gaps, the Steiner Tree optimization recovers the minimum connected subgraph, identifying intermediate tables required for valid 4+ table joins. These results indicate that enforcing topological constraints is critical for robust schema linking in complex databases. We further verify Graph-Link's scalability on the 65-table `works_cycles` database: it achieves a Hit Rate of 88.3% and an EX of 70.5%, substantially outperforming MCS-SQL (74.6% Hit, 61.3% EX). Detailed results are provided in Appendix A.7.

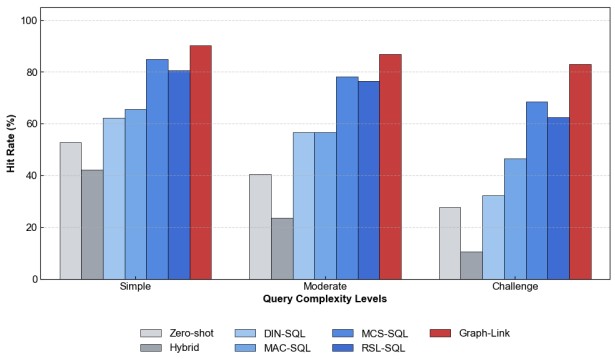

*Figure 3.* **Hit Robustness Across Query Complexity.**

### 5.4. End-to-End SQL Generation

To isolate the impact of schema linking on final SQL generation, we decouple retrieval from generation and evaluate all methods under a standardized *Fixed Generator*. The

*Table 4.* End-to-End Execution Accuracy (EX) and Efficiency on BIRD Dev.

| Linking Method | EX (All) | EX (Simp.) | EX (Mod.) | EX (Chal.) | Avg. Tokens |
|---|---|---|---|---|---|
| ***Baselines*** | | | | | |
| Full Schema (No Link) | 45.7% | 52.4% | 40.3% | 21.5% | ~55,300 |
| Hybrid (BM25+Vector) | 29.4% | 38.4% | 18.6% | 6.9% | ~8,500 |
| DIN-SQL (Linker Only) | 52.4% | 58.3% | 48.1% | 28.9% | ~33,200 |
| MAC-SQL (Linker Only) | 54.8% | 59.6% | 52.3% | 32.9% | ~15,800 |
| MCS-SQL (Linker Only) | 66.0% | 72.8% | 59.3% | 44.8% | ~63,800 |
| RSL-SQL (Linker Only) | 63.4% | 69.6% | 58.4% | 40.8% | ~48,100 |
| ***Ours*** | | | | | |
| **Graph-Link** | **68.5%** | **73.3%** | **62.2%** | **58.6%** | ~20,800 |
| ***Oracle Upper Bound*** | | | | | |
| *Golden Link* | 72.6% | 76.0% | 67.9% | 66.2% | ~6,800 |

pipeline is model-agnostic and consists of schema serialization (Pourreza & Rafiei, 2023), locality-sensitive value retrieval (Deng et al., 2025), CoT-based reasoning (Wei et al., 2022), and single-turn execution-guided correction (Tian et al., 2024).

As shown in Table 4, Graph-Link achieves an EX of **68.5%** on BIRD Dev, outperforming the strongest agentic baseline, MCS-SQL, by 2.5 points and the Full Schema baseline by over 20 points. The gains are most pronounced on complex queries: on the *Challenge* subset, Graph-Link reaches **58.6%** EX, compared to 44.8% for MCS-SQL and 32.9% for MAC-SQL. This improvement reflects Graph-Link's ability to recover implicit bridge tables via Steiner-tree optimization, thereby avoiding invalid join paths commonly induced by purely semantic retrieval. In addition, Graph-Link substantially reduces prompt noise, using only ~20,800 tokens on average versus ~63,800 for MCS-SQL.

### 5.5. Ablation Studies

To validate the effectiveness of the proposed modules in Graph-Link, we conducted ablation studies on the BIRD development set by systematically removing individual components. Table 5 presents the results, comparing the full framework against variants without Hierarchical Community (HC), Dual-Perception (DP), Steiner Tree (ST), and LLM-Critic (LC). The results demonstrate that removing any module negatively impacts performance. The **Steiner Tree** backbone proves most critical: its removal results in a catastrophic 11.2% drop in Hit Rate, confirming that topological pathfinding is essential for recovering implicit bridge tables that semantic retrieval misses. Additionally, the **LLM-Critic** plays a vital role in noise filtration, as evidenced by the sharp 13.1% decline in Precision when removed.

### 5.6. Sensitivity Analysis

We further examine the sensitivity of Graph-Link to its four most influential hyperparameters: the anchor threshold $\delta_{anchor}$, the community routing threshold $\tau_{route}$, the dual-scale weight $\alpha$, and the column pruning threshold $\delta_{prune}$.

*Table 5.* **Ablation Study of Graph-Link on BIRD-Dev.**

| Model Variant | Precision | Recall | F2 Score | Hit Rate |
|---|---|---|---|---|
| **Graph-Link (Full)** | **70.4** | **95.9** | **89.4** | **90.8** |
| w/o HC | 69.7 (-0.7) | 92.4 (-3.5) | 86.6 (-2.8) | 86.5 (-4.3) |
| w/o DP | 68.0 (-2.4) | 92.7 (-3.2) | 86.3 (-3.1) | 87.2 (-3.6) |
| w/o ST | 75.5 (+5.1) | 87.5 (-8.4) | 83.7 (-5.7) | 79.6 (-11.2) |
| w/o LC | 57.3 (-13.1) | **96.4** (+0.5) | 85.9 (-3.5) | 90.3 (-0.5) |

As shown in Figure 4, Graph-Link exhibits strong stability across practical ranges. Varying $\delta_{anchor}$ between 0.55 and 0.75 leads to a Hit Rate change of less than $\pm1.5\%$, while $\tau_{route}$ and $\alpha$ each induce variations within $\pm2.3\%$. The pruning threshold $\delta_{prune}$ primarily affects precision, with a variation of $\pm3.0\%$, and leaves recall nearly unchanged ($\pm1.8\%$). These results confirm that Graph-Link does not rely on brittle parameter tuning, and the optimal region is broad and flat.

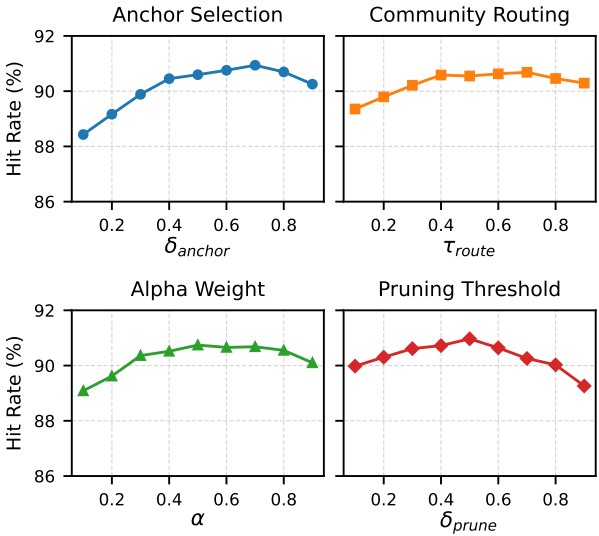

*Figure 4.* Sensitivity of Hit Rate on BIRD-Dev.

# 6. Conclusion

We propose **Graph-Link** to resolve *structural blindness* by reformulating schema linking as topology-constrained subgraph induction. By synergizing semantic retrieval with Steiner-tree optimization, our framework is the first to explicitly enforce global connectivity as a hard constraint during schema selection, shifting the paradigm from independent ranking to connected subgraph recovery. Extensive experiments on BIRD and Spider 2.0 demonstrate that Graph-Link achieves state-of-the-art schema completeness, raising the Hit Rate to 90.8% and improving downstream execution accuracy on complex multi-hop queries by 13.8%. These gains are particularly pronounced in challenge scenarios, where the recovery of implicit bridge tables is critical.

Our findings confirm that explicit structural reasoning is not merely beneficial but necessary for robust, real-world Text-to-SQL systems, and we hope that Graph-Link can serve as a foundation for future research in topology-aware database interfaces.

# Limitations

Graph-Link relies on explicit foreign key constraints, potentially limiting performance on denormalized or weakly structured schemas. Additionally, graph traversal introduces computational overhead compared to pure retrieval methods. Finally, our focus remains on structural connectivity; resolving semantic ambiguity in underspecified user queries requires complementary future work.

# Future Work

While Graph-Link establishes strong structural guarantees, several directions remain open. First, evaluating on truly large-scale databases (100+ tables) with highly complex multi-hop queries is essential to fully validate scalability, but is currently limited by the absence of public benchmarks; constructing such a benchmark is an important community goal. Second, adapting the hierarchical index to frequent schema changes via incremental community re-partitioning would reduce offline maintenance cost in dynamic enterprise environments. Third, incorporating query execution feedback to refine the Steiner-tree resistance weights could close the loop between structural connectivity and semantic correctness, further improving precision without sacrificing recall. Finally, extending the framework to support multi-turn and conversational Text-to-SQL settings, where context accumulation demands dynamic subgraph expansion, presents a promising avenue for practical deployment.

# Acknowledgements

This work is supported by National Key R&D Program of China (2023YFB2904100).

# Impact Statement

While our approach enhances the structural faithfulness of SQL generation, we acknowledge the potential societal consequences of deploying automated database interfaces. Specifically, the democratization of data access necessitates rigorous data governance to ensure user privacy and data security. We emphasize that such systems should operate within strict permission boundaries to prevent unauthorized information retrieval, and critical decision-making based on generated queries should involve human verification. There are no other major ethical concerns that we feel must be specifically highlighted here.

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

# A. Implementation Details

In this section, we provide comprehensive details regarding the benchmarks, hyperparameters, and baseline configurations to ensure the reproducibility of Graph-Link.

## A.1. Benchmarks and Data Statistics

We evaluate Graph-Link on two premier cross-domain benchmarks that represent the current state-of-the-art in Text-to-SQL difficulty. These datasets were selected specifically because they exhibit high schema complexity and require significant multi-hop reasoning, making them ideal testbeds for topological subgraph induction.

**BIRD: Big Relational Database Benchmark.**   Li et al. (2023b) is currently the most challenging large-scale cross-domain dataset. Unlike previous benchmarks (e.g., Spider 1.0), BIRD focuses on bridging the gap between academic research and real-world applications by introducing two major challenges: **dirty database values** and **massive schema sizes**.

- **Scale and Complexity:** BIRD contains 12,751 unique question-SQL pairs across 95 databases. The databases are significantly larger, with a total size of 33.4 GB.

- **External Knowledge:** A unique feature of BIRD is the inclusion of "External Knowledge" (evidence) for queries, requiring models to link ambiguous natural language terms to database values.

- **Relevance to Graph-Link:** We specifically utilize the **Dev set** (1,534 examples) for evaluation. BIRD is critical for our analysis because its databases often contain dozens of tables (avg. 15.6 tables/DB) with non-obvious foreign key paths, heavily penalizing methods that suffer from *Structural Blindness*.

**Spider 2.0.**   Lei et al. (2024) is the successor to the classic Spider benchmark, designed to simulate enterprise-grade data analytics problems. It moves beyond simple translation tasks to address challenges in **SaaS-like environments**, including diverse SQL dialects (BigQuery, Snowflake), data privacy constraints, and schema evolution.

- **Spider 2.0-Lite:** Due to the prohibitive cost and latency of evaluating on the full multi-modal Spider 2.0 suite, we utilize the **Spider 2.0-Lite** subset for schema linking evaluation. This subset preserves the high structural complexity (large schema graphs) of the full version while removing multimodal constraints (e.g., image-based queries), focusing strictly on the text-to-structure grounding capability.

- **Topological Challenge:** Spider 2.0 schemas feature a higher degree of normalization and node connectivity compared to Spider 1.0, requiring the retrieval of extended join paths (3+ hops) which Graph-Link is designed to optimize.

**Data Statistics Summary.** Table 6 provides a statistical comparison of the benchmarks. Notably, the "Avg. Tables per DB" and "Avg. Joins per Gold SQL" metrics in BIRD and Spider 2.0 are significantly higher than in legacy datasets, confirming the necessity for a graph-based linking approach.

*Table 6.* Statistics of the benchmarks used in evaluation. BIRD and Spider 2.0 feature significantly larger search spaces compared to legacy datasets.

| Statistic | Spider 1.0 | BIRD (Dev) | Spider 2.0-Lite |
|---|---|---|---|
| # Databases | 166 | 11 | 45 |
| # Question-SQL Pairs | 10,181 | 1,534 | 820 |
| Avg. Tables / DB | 5.3 | **15.6** | 12.4 |
| Avg. Columns / DB | 27.6 | **142.3** | 88.5 |
| Avg. Joins / Query | 1.8 | 2.6 | **2.9** |
| Schema Size (Tokens) | ∼4k | ∼28k | ∼19k |

### A.2. Graph Construction (Offline Phase)

**Embeddings and Similarity.** For node initialization, we utilized `text-embedding-v4` (dimension $d = 1024$) to generate dense vector representations.

- **Table Embeddings ($\mathbf{g}_t$):** Computed by concatenating the table name and its description (summarized by LLMs).

- **Column Embeddings ($\mathbf{h}_c$):** Computed using the format `Table Name.Column Name: Column Description`.

To compute the **Dual-Scale Similarity** (Eq. 2), we set the balancing factor $\alpha = 0.6$, giving slightly more weight to table-level semantic alignment than individual column matching during edge creation. The similarity pruning threshold $\tau$ was set to **0.75**; edges with cosine similarity below this value were pruned to maintain graph sparsity, except for explicit Foreign Key (FK) edges which are always preserved.

**Community Partitioning.** We applied the Leiden algorithm with a **resolution parameter of 1.0** and ran it for 5 iterations to ensure stable convergence of the modularity $Q$. For the BIRD dataset, this resulted in an average of $4.2$ communities per database.

### A.3. Steiner Tree Optimization (Online Phase)

**Approximation Algorithm.** Solving the Node-Weighted Steiner Tree (NWST) is NP-hard. We implemented the 2-approximation algorithm based on the metric closure on the induced subgraph $\mathcal{G}_{view}$.

1. **Metric Closure:** We computed all-pairs shortest paths between Anchor Nodes ($\mathcal{A}$) using Dijkstra's algorithm (weighted by edge resistance).

2. **MST Construction:** We constructed the Minimum Spanning Tree on the metric closure using Kruskal's algorithm.

3. **Path Unfolding:** Abstract edges in the MST were replaced by their corresponding physical shortest paths in the original graph to form the Steiner backbone.

To prevent infinite loops in cyclic schemas, we enforced a **maximum path length of 6 hops** when computing distances in the metric closure.

### A.4. LLM Configuration and Inference

For all generation and reasoning tasks (Query Decomposition, LLM-Critic), we utilized **Qwen3-max** and **GLM-4.7** via API.

- **Temperature:** Set to $0.0$ to ensure deterministic outputs.

- **Top-P:** Set to $1.0$.

- **Max New Tokens:** Limited to 1024 for SQL generation and 512 for reasoning steps.

- **Instruction Following:** We utilized strictly formatted system prompts to force the LLM to output valid JSON for the Query Decomposition phase, ensuring parsability.

### A.5. Hyperparameter Settings

Table 7 details the specific hyperparameter values used across all reported experiments. These values were determined via grid search on the BIRD development set (using a 20% hold-out split).

### A.6. Baseline Implementation Details

To ensure fair comparison, all baselines were reproduced using their official open-source repositories with the following standardizations:

*Table 7.* Hyperparameter settings for Graph-Link.

| Parameter | Symbol | Value |
|---|---|---|
| *Graph Structure* | | |
| Dual-Scale Weight | $\alpha$ | 0.6 |
| Edge Pruning Threshold | $\tau$ | 0.75 |
| Structural Edge Weight | $\epsilon$ | $1e^{-4}$ |
| *Anchor Localization* | | |
| Community Selection Threshold | $\tau_{\text{route}}$ | 0.6 |
| Topological Prior Strength | $\lambda$ | 0.5 |
| Anchor Selection Threshold | $\delta_{anchor}$ | 0.65 |
| *Context Optimization* | | |
| Column Pruning Threshold | $\delta_{prune}$ | 0.45 |
| Max Hops (Steiner) | $H_{max}$ | 6 |

- **Hybrid Retrieval:** Implemented using `Elasticsearch` for BM25 and `FAISS` for vector retrieval, with a fusion weight of 0.3 (BM25) and 0.7 (Dense).

- **DIN-SQL:** We used the "schema linking" module from the official implementation, limiting the number of retrieved columns to 100 to fit the context window.

- **MAC-SQL:** We adopted the standard multi-agent collaboration setting (Selector and Refiner agents) and restricted the maximum dialogue turns to 3 to balance reasoning depth with inference latency.

- **MCS-SQL:** We set the sampling size $N = 5$ for the multiple-choice selection phase with a temperature of 0.7 to ensure reasoning diversity, utilizing majority voting to determine the final schema consensus.

- **RSL-SQL:** We configured the max retrieval depth to 3 and beam size to 5, consistent with their reported best settings.

### A.7. Scalability on Larger Schemas

To assess performance under larger databases, we evaluate on `works_cycles` (65 tables, BIRD training set). Table 8 reports the linking and end-to-end execution accuracy.

*Table 8.* Performance on `works_cycles` (65 tables).

| Method | Hit Rate (%) | EX (%) |
|---|---|---|
| Hybrid (BM25+Vector) | 53.8 | 38.2 |
| MAC-SQL (linker only) | 69.2 | 54.6 |
| MCS-SQL | 74.6 | 61.3 |
| **Graph-Link** | **88.3** | **70.5** |

## B. Case Study

We present a qualitative case study to illustrate how Graph-Link performs hierarchical schema construction, topology-aware schema linking, and downstream SQL generation in a realistic multi-table enterprise database. We specifically highlight how the framework recovers implicit "bridge tables" that pure semantic retrieval methods typically miss.

### B.1. Database Scenario

Consider an enterprise sales database consisting of three loosely coupled subsystems. To optimize space and readability, we denote primary keys with an underline (pk) and foreign keys with a dagger ($^{\dagger}$).

**Core Sales Community (Target Domain).** The analytical focus is on revenue and customer behavior. The schema includes:

- customers($\underline{cust\_id}$, name, region)
- orders($\underline{order\_id}$, cust_id$^\dagger$, date)
- order_items($\underline{item\_id}$, order_id$^\dagger$, prod_id$^\dagger$, qty)
- products($\underline{prod\_id}$, cat_id$^\dagger$, price)
- categories($\underline{cat\_id}$, cat_name)
- payments($\underline{pay\_id}$, order_id$^\dagger$, amount)

**Logistics Community (Irrelevant Context).** Structurally connected via orders but semantically orthogonal:

- warehouses(...), shipments(...), carriers(...)

Standard foreign key constraints connect these entities as depicted in Figure 5. Although the shipments table connects to orders, it represents a semantic drift from the revenue-focused query and should be pruned.

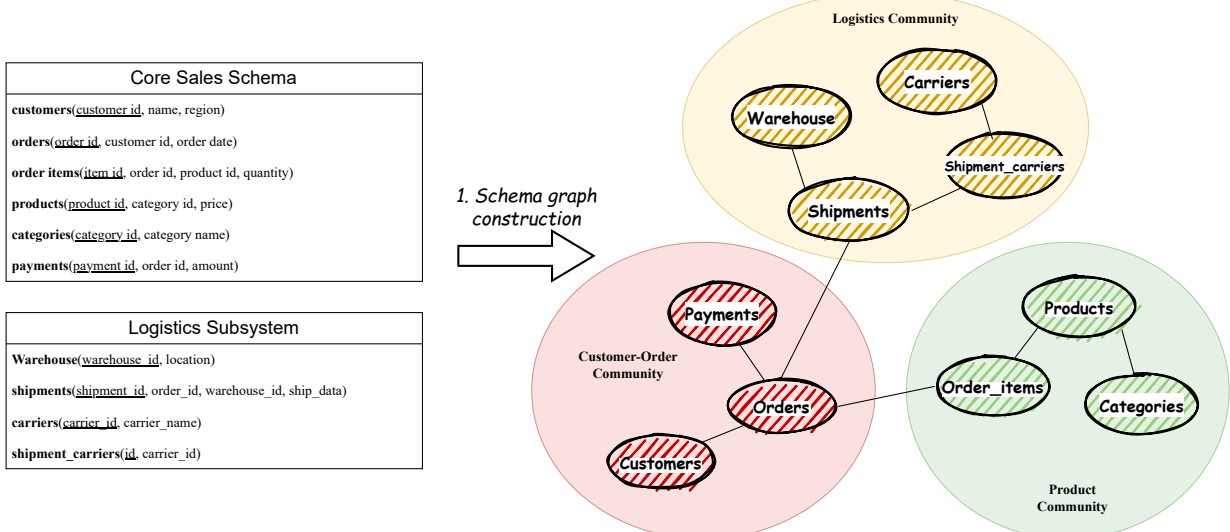

*Figure 5.* Hierarchical Schema Graph Construction showing the partitioning of Core Sales vs. Logistics communities.

### B.2. Dual-Perception: Decomposition & Anchor Localization

The input query is: *"What is the total revenue from electronics purchased by customers in Europe in 2023?"*

Graph-Link first decomposes this query into atomic intents and maps them to semantic anchors. As shown in Table 9, purely semantic retrieval identifies customers, categories, and payments, but fails to identify orders and order_items because the query lacks keywords like "order" or "item".

### B.3. Topology-Aware Subgraph Induction

To generate valid SQL, the disjoint anchors identified above must be connected. Graph-Link solves the Steiner Tree problem on the schema graph, inducing a minimal connected subgraph.

The optimized topological backbone recovers the following path, where **Bold** indicates anchors and *Italics* indicates induced bridge tables:

1. **customers** $\xrightarrow{\text{FK}}$ *orders* (*Bridge 1*: Connects Customer to Transaction)

*Table 9.* Mapping decomposed sub-queries to schema anchors. Note that structural bridges (`orders`, `order_items`) remain unidentified at this stage.

| Sub-query Intent | Key Semantic Terms | Identified Anchor |
|---|---|---|
| 1. Customer Filtering | "customers", "Europe" | **customers** |
| 2. Category Selection | "electronics" | **categories** |
| 3. Revenue Aggregation | "total revenue" | **payments** |
| 4. Temporal Constraint | "2023" | *None (Missed)* |

2. $orders \xrightarrow{\text{FK}}$ **payments** (Links Revenue)

3. $orders \xrightarrow{\text{FK}} order\_items$ (*Bridge 2*: Connects Transaction to Product)

4. $order\_items \xrightarrow{\text{FK}} products \xrightarrow{\text{FK}}$ **categories**

Notably, the $orders$ table is recovered purely based on topological centrality, despite having low semantic similarity to the prompt. Simultaneously, the Steiner Tree algorithm avoids expanding into the `Logistics` community (e.g., `shipments`), as doing so would increase the total path cost without connecting to new anchors.

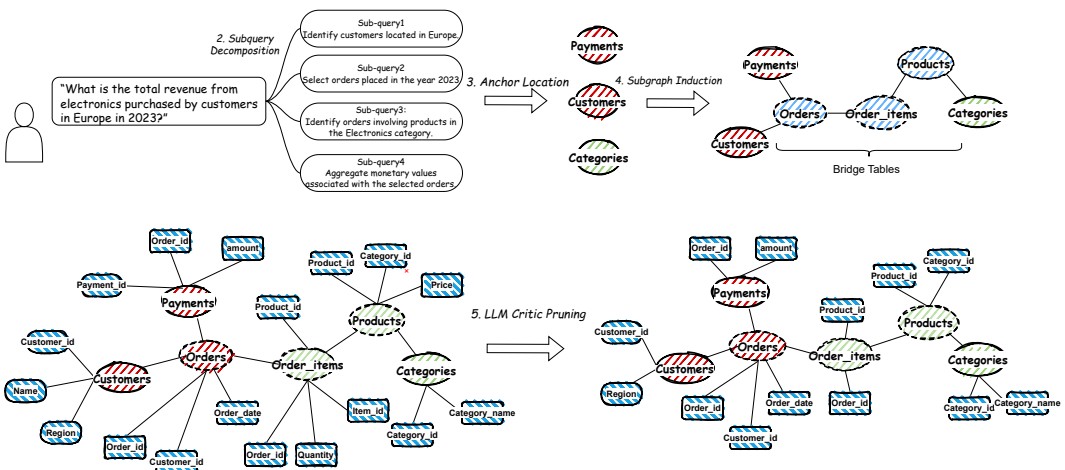

*Figure 6.* The induced Steiner Tree. Red nodes are semantic anchors; Blue nodes are structurally recovered bridge tables.

### B.4. Downstream SQL Generation

The pruned schema is passed to the SQL generator. The resulting query demonstrates correct multi-hop joining capabilities, as highlighted by the annotations:

```sql
SELECT SUM(p.amount) AS total_revenue
FROM customers c
-- Bridge Table 1: Recovered via Topology
JOIN orders o ON c.customer_id = o.customer_id
-- Target Anchor: Revenue
JOIN payments p ON o.order_id = p.order_id
-- Bridge Table 2: Recovered via Topology
JOIN order_items oi ON o.order_id = oi.order_id
JOIN products pr ON oi.product_id = pr.product_id
JOIN categories cat ON pr.category_id = cat.category_id
WHERE c.region = 'Europe'
  AND cat.category_name = 'Electronics'
  AND strftime('%Y', o.order_date) = '2023';
```

### B.5. Comparison with Baselines

This case illustrates the critical advantage of Graph-Link over standard retrieval:

**Baseline Failure Mode:** A standard dense retriever (e.g., Hybrid or Zero-shot) would retrieve `customers` and `payments` but likely prune `orders` and `order_items` due to low semantic overlap. This results in the "Missing Bridge" problem, leading to an un-executable SQL query (e.g., trying to join `customers` directly to `payments`). **Graph-Link Success:** By decoupling semantic relevance (Anchors) from structural necessity (Steiner Tree), our method guarantees topological completeness, ensuring that the generated SQL contains valid join paths for complex multi-hop reasoning.

## C. Prompt Templates

### C.1. Inter-Table Semantic Relationship Analysis

---

**Inter-Table Semantic Relationship Analysis Prompt**

```
You are a senior data warehouse architect and semantic data modeling expert. Your task
    is to analyze raw database schema metadata and produce a comprehensive, semantic
    understanding of both individual tables and inter-table relationships, from a
    business-oriented perspective rather than a purely technical one.

You must reason step-by-step internally, but **only output the final structured
    results**.

### 1. Table-Level Semantic Summarization
For each table, infer its business meaning and structural role.

### 2. Inter-Table Semantic Relationship Discovery
Based on the inferred table semantics and column information, identify all meaningful
    relationships between tables.

## INPUT
schema context are as followed:
{SCHEMA CONTEXT}

## Output Format (Strict JSON)

```json
{
  "table_summaries": [
    {
      "table_name": "TableA",
      "description": "One-sentence description of the table's core business purpose.",
      "granularity": "What a single row represents at the business level.",
      "key_concepts": ["List of inferred business concepts derived from column semantics
          "],
      "potential_keys": ["List of possible primary keys or unique identifiers"]
    }
  ],
  "relationships": [
    {
      "source_table": "TableA",
      "target_table": "TableB",
      "relationship_type": "EXPLICIT_JOIN | SEMANTIC_LINK",
      "join_condition": "TableA.col = TableB.col",
      "confidence": 0.0,
      "description": "Brief explanation of the semantic relationship"
    }
  ]
}
```

## C.2. Community Summarization Prompt

**Schema Community Semantic Summarization Prompt**

```
You are a **Domain-Driven Design (DDD) Expert** and **Data Warehouse Architect**. Your
    task is to analyze a specific "community" of tables a cluster of tightly connected
    tables detected by graph algorithms (e.g., Louvain or Leiden) within a database
    schema graph.

You must reason step-by-step internally to identify the **Core Business Domain** (e.g.,
    "Order Processing", "Inventory Management"), but **only output the final
    structured results**.

## Objectives
Based on the provided nodes and edges, you must:

### 1. Identify Core Components
Determine which tables serve as the "hubs" or "protagonists" within this cluster.

### 2. Abstract Business Themes
Analyze what specific business problem or data entity these tables collectively
    address to generate a high-level business summary.

### 3. Analyze Cohesion
Explain the structural or semantic reasons why these tables are clustered together.

## INPUT
The community context is as follows:

### 1. Community Table Nodes
{NODES_LIST}

### 2. Community Relationships (Edges)
{EDGES_LIST}

## Output Format (Strict JSON)

```json
{
  "community_name": "Short business domain name (3-5 words)",
  "summary": "Detailed description explaining what business function these tables
      collectively fulfill.",
  "core_tables": ["List of identified central/hub tables in the community"],
  "functional_tags": ["List of key business actions supported, e.g., 'Payment
      Processing', 'User Tracking'"],
  "semantic_cohesion": "Explanation of why these tables are clustered (e.g., 'Tightly
      coupled via foreign keys', 'Shared business keys', 'Semantic similarity')"
}
```

## C.3. Query Decomposition

---

**Schema-Guided Query Decomposition Prompt**

```
# Schema-Guided Query Decomposition Prompt
You are a Database Query Planner. Your task is to break down a complex User Query into
    a series of logically related Sub-queries and assign the most appropriate Schema
    Community to each sub-query.

### Core Logic
1. **Analyze Intent**: Understand the user's ultimate goal and identify if the query
    involves Cross-domain or Multi-hop logic.
2. **Community Matching**: Based on the provided "Available Schema Communities",
    determine which set of tables contains the information needed to answer specific
    parts of the question.
3. **Logical Decomposition**:
   - If the question involves multiple communities (e.g., finding employees in "HR
      Community" then checking performance in "Sales Community"), you must decompose
      it.
   - You must maintain dependencies between sub-questions (e.g., the output of Step 1
      serves as a filter for Step 2).
4. **No Decomposition Needed**: If the question is simple and involves only one
    community, generate a single step.

## Input
### User Query
{USER_QUERU}

### Backgroud Knowledge
{BACKGROUND_KNOWLEDGE}

###Database Community Introdution
{COMMUNITY SUMMARY}

### Output Format (Strict JSON)
You must reason step-by-step internally, but return strictly a JSON object formatted
    as follows:

```json
{
  "decompostions":["sub_query1", "sub_query2",]
}
```

---

## C.4. Subgraph Pruning Prompt

**Prompt used for Schema Pruning and Column Selection**

```
As an experienced and professional database administrator, your task is to analyze a
    user question and a schema subgraph to provide relevant information.
Your goal is to identify the relevant tables and columns based on the user question
    and evidence provided.

[Instruction]:
1. Discard any table schema that is not related to the user question and evidence.
2. Select the most relevant columns in each relevant table. Discard irrelevant columns.

3. Ensure that sufficient tables are included in the final output to answer the
    question, including tables needed for joining (Foreign Keys).

[Recall-Oriented Strategies – CRITICAL]:
1. **MAXIMUM COVERAGE PRINCIPLE**: When in doubt, INCLUDE additional columns that
    might be relevant. It's better to have extra columns than miss critical ones.
2. **JOIN PATH PRESERVATION**: Always include ALL foreign key columns and their
    corresponding primary key columns to maintain complete join paths.
3. **PRIMARY KEY INCLUSION**: ALWAYS include primary key columns (usually 'id') for
    every selected table – they are essential for:
  – Identifying and returning specific entities (e.g., "which cards", "what students")
  – Joining tables correctly
  – Ensuring query correctness
4. **Ambiguity Handling**: If the user question can be interpreted in multiple ways or
     involves multiple potential join paths, select schema elements for ALL
    possibilities.
5. **Metadata Retention**: Include descriptive columns (names, labels, identifiers)
    alongside technical columns to provide context for query construction.
6. **Multi-hop Dependency**: Consider indirect relationships and include intermediate
    tables/columns that enable complex joins.

## INPUT
### User query
{USER_QUERY}

### Schema subgraph context
{SUBGRAPH_CONTEXT}

### OUTPUT FORMAT :
```json
{
  "selected_schema": {
    "table_name": ["col1", "col2"]
  },
  "foreign_keys": ["table1.column = table2.column"],
  "filter_conditions": [
      "column = value"
   ],
  "is_sufficient": boolean,
  "missing_info": "Description of missing info if any"
}
```

## C.5. SQL Generation

> **Prompt of Final SQL Generation**
>
> ```
> You are an expert SQL Data Analyst with strong reasoning capabilities. Your task is to
>     generate a correct SQLite-compatible SQL query by following a step-by-step
>    reasoning process.
> ## Reasoning Process (Chain of Thought)
> - Understand the Question: Break down what the user is asking.
> - Identify Sub-problems: Can this be solved by answering simpler sub-questions?
> - Plan the Query Structure: Determine required JOINs, filters, aggregations, and
>     ordering.
> - Construct SQL: Write the query using schema_links for JOIN conditions.
>
> ## Few-Shot Examples (Learn the Reasoning Pattern)
> ### Example 1: Q: "Find the title of courses that have two prerequisites?"
>     Schema_links: [course.title, course.course_id = prereq.course_id] A: Let's think
>     step by step. "Find the title of courses that have two prerequisites?" can be
>     solved by knowing the answer to the following sub-question "What are the titles for
>      courses with two prerequisites?". The SQL query for the sub-question "What are the
>      titles for courses with two prerequisites?" needs to:
>
> - JOIN course and prereq tables on course_id
> - GROUP BY course to count prerequisites
> - Filter courses with exactly 2 prerequisites using HAVING SQL: SELECT T1.title FROM
>     course AS T1 JOIN prereq AS T2 ON T1.course_id = T2.course_id GROUP BY T2.course_id
>      HAVING count(*) = 2
>
> ## CRITICAL SQL Best Practices
> 1. **Use Schema Links**: Leverage provided foreign_keys for JOIN conditions.
> 2. **Use Filter Conditions**: Apply provided filter_conditions when available.
> 3. **NULL Handling**: Filter NULL values before ORDER BY, MAX, MIN operations.
> 4. **Column Type Priority**: Prefer categorical columns over LIKE matching.
> 5. **Exact Names**: Use ONLY the original table/column names from Retrieved Schema.
> 6. **SELECT Column Strategy**:
> * Analyze what information the question asks for.
> * If the question asks 'which/what entities', return the PRIMARY KEY (usually 'id').
> * If specific attributes are requested, select ONLY those columns.
> * AVOID 'SELECT *' unless the question explicitly asks for 'all information'.
> * Default to minimal necessary columns for efficiency.
>
> ## INPUT
> User Question: {QUESTION}
> Evidence / Hint: {EVIDENCE}
> Retrieved Schema: {SCHEMA_CONTEXT}
> Schema Links (Join Hints & Filter Conditions)
> **Foreign Keys (for JOINs)**: [{FOREIGN_KEYS}]
> **Filter Conditions**: [{FILTER_CONDITIONS}]
> Value Exploration Results: {EXPLORATION_CONTEXT_STR}
> ## Output Format
> Provide your response in the following format:
> **Reasoning:**
> Let's think step by step. [Your detailed reasoning process here]
>
> **Sub-question (Optional):**
> [If applicable, state the simplified sub-question]
>
> **SQL:**
>
> ```sql
> SELECT ... (final SQLite query)
> ```

