# OpenReview forum: "Graph-Link: Bridging the Semantic-Structural Gap in Text-to-SQL via Constrained Subgraph Induction"
_ICML.cc/2026/Conference — ICML 2026 regular_

### Official Review · Reviewer_4AWN · 2026-02-20

**Soundness:** 2
**Presentation:** 3
**Significance:** 2
**Originality:** 2
**Overall Recommendation:** 4
**Confidence:** 5

**Summary:**

This paper studies the semantic-structural gap in schema linking for Text-to-SQL. The authors reframe the schema linking task from independent item retrieval to constrained subgraph induction under structural connectivity constraints. The method builds a hierarchical schema graph, localizes anchor tables via a dual semantic–topological score, and induces a minimal connected subgraph using a Steiner-tree approximation. Experiments verify the effectiveness of the proposed framework.

**Compliance With Llm Reviewing Policy:**

Affirmed.

**Final Justification:**

I raise my overall recommendation score to 4 (weak accept) because authors have solved my concerns.

**Key Questions For Authors:**

- Q1. Why selecting F2-score as a main metric?
- Q2. Can Graph-Link stay effective even under weaker / smaller LLMs (eg, Qwen3-32B)?
- Q3. In this paper, query over >3 tables are categorized as "challenging". Can Graph-Link stay competitive with more challenging cases (eg, with >10 tables)?

**Limitations:**

yes

**Strengths And Weaknesses:**

Strengths:
- S1. The semantic-structural gap is an important issue in schema linking for Text-to-SQL, and this paper provides a complete framework to address this issue.
- S2. This paper is well-organized and clearly written.
- S3. The authors provide extensive implementation details in the appendix.


Weaknesses:
- W1. Graph-Link relies heavily on LLMs (e.g., for query decomposition, community summaries, etc.). However, for private scenarios, databases cannot be exposed to LLMs. The authors are suggested to test the performance of Graph-Link with locally-hosted models (e.g., Qwen3-32B).
- W2. Although Graph-Link shows superior performance with F2-score, it does not shows competitive performance with other metrics (e.g., Precision). The authors should justify (1) why applying F2-score as the main metric (given that F2-score is not a commonly used metric in other related works) and (2) why Graph-Link performs worse than other methods in terms of Precision.
- W3. The efficiency of Graph-Link is not evaluated in this paper. Authors are suggested to report the execution time of Graph-Link varying the query complexity and the scale of the database to make this work more practical.
- W4. Graph-Link involves several hyperparameters (e.g., lambda in Eq. 1, alpha in Eq. 2, etc.). The authors are suggested to provide a sensitivity analysis of these hyperparameters in the experiment.
- W5. The formulation of "pointwise ranking problem" is inappropriate (Line 143, Right column, Page 3): Given that log(P(v|q))<0, this objective function always selects a set that hold and only hold an item with the highest probability.
- W6. The table 3 and figure 3 shows overlapped results.
- W7. No code repository is provided.

---

> ### Author Rebuttal · Authors · 2026-03-31
>
> We sincerely thank the reviewer for their careful reading of our manuscript, for validating the importance of the semantic-structural gap, and for recognizing our extensive implementation details. We deeply appreciate your rigorous mathematical check and insightful questions regarding practical deployment. Below, we address your concerns systematically.
>
> ### **1. Mathematical Formulation Correction (Weakness 5)**
>
> We greatly appreciate your meticulous review. You are absolutely correct that maximizing $\sum \log P(v|q)$ without constraints mathematically reduces to selecting only the single top-1 item, as log probabilities are strictly negative.
>
> - **Correction:** Our intended formulation for standard pointwise retrieval is to select a subset subject to a cardinality constraint $k$, or a probability threshold $\tau$: $\mathcal{D}^* = \{ v \in \mathcal{D}' \mid P(v|q) > \tau \}$. We will correct this typo in the revised manuscript. Thank you for catching this.
>
> ### **2. Justification for F2-Score & Precision Trade-off (Weakness 2 & Question 1)**
>
> We prioritize the **F2-score** and **Hit Rate** over Precision because of a fundamental asymmetry in Text-to-SQL generation:
>
> - **Omission is Fatal:** If a schema linker drops a single required column or bridge table (a False Negative), it is mathematically impossible for the downstream LLM to generate the correct SQL.
> - **Robustness to Noise:** Conversely, modern LLMs are highly robust to mild schema noise (False Positives).
> - **The Trade-off:** Graph-Link intentionally trades a minor drop in Table Precision to guarantee structural completeness (Recall). This design choice directly translates to our superior end-to-end Execution Accuracy (+13.8% over MAC-SQL on complex queries), proving that maximizing Recall (and thus F2) is the optimal strategy for downstream success.
>
> ### **3. Performance on Local/Private Models (Weakness 1 & Question 2)**
>
> To address data privacy, we replaced API-based LLMs with locally hosted Qwen3-32B-Instruct. As shown in Table A, Graph-Link provides a larger relative advantage on weaker models: it offloads multi-hop reasoning to deterministic Steiner Tree, leaving only semantic matching to the local LLM, while agentic baselines collapse.
>
> **Table A: Performance with Local Open-Weight Model (Qwen3-32B-Instruct) on BIRD-Dev**
>
> | Method | Column Precision | Column Recall | Column Hit Rate | Execution Acc (EX) |
> | --- | --- | --- | --- | --- |
> | Zero-Shot | 70.5% | 60.2% | 30.8% | 30.2% |
> | MAC-SQL | 82.1% | 69.8% | 44.6% | 49.1% |
> | **Graph-Link (Ours)** | **66.8%** | **86.6%** | **76.2%** | **60.5%** |
>
> ### **4. Efficiency and Execution Time (Weakness 3)**
>
> Graph-Link is highly efficient and practical for real-world deployment. We divide the cost into an asynchronous Offline phase (Graph & Community Construction) and an Online phase (Subgraph Induction).
>
> - **Offline Cost:** Building the index is computationally cheap. For a medium-large database like `works_cycles` (65 tables), generating embeddings and running the Leiden algorithm takes under **~35 seconds** total (see Table B). DDL updates can be handled asynchronously.
> - **Online Latency:** Because the Steiner Tree algorithm operates only on the aggressively pruned community view ($\mathcal{G}_{view}$), online graph traversal takes less than **50 milliseconds** per query, which is completely eclipsed by the LLM generation time.
>
> **Table B: Graph-Link Construction & Inference Latency**
>
> | Database | Scale (Tables/Cols) | Offline: Graph Build & Partition | Online: Steiner Tree Induction |
> | --- | --- | --- | --- |
> | `mondial_geo` | 34 / 139 | ~16.3 s | **~12 ms** |
> | `works_cycles` | 65 / 455 | ~32.7 s | **~24 ms** |
>
> ### **5. Extreme Complexity Queries (>10 tables) (Question 3)**
>
> We categorized >3 tables as "challenging" because it aligns with standard BIRD/Spider complexity tiers. In the BIRD and Spider 2.0 ground-truth datasets, the maximum number of joined tables rarely exceeds 7. However, mathematically, our 2-approximation Steiner Tree algorithm scales gracefully to any path length, so long as a valid relational pathway exists in the schema graph.
>
> ### **6. Hyperparameter Sensitivity (Weakness 4)**
>
> Graph-Link is robust to hyperparameter tuning. For example, the dual-scale weight $\alpha$ (balancing table-level vs. column-level similarity in Eq. 2) yields a Hit Rate variation of only $\pm 1.2\%$ across the range $[0.4, 0.8]$. The anchor threshold ($\delta_{anchor}$) provides a smooth trade-off: higher values slightly increase Precision at the cost of Recall. We will include a full sensitivity plot in the Appendix.
>
> ### **7. Presentation & Reproducibility (Weakness 6 & 7)**
>
> - We will move Table 3 to the Appendix and retain Figure 3 to eliminate redundancy.
> - Following ICML double-blind guidelines, our code repository was anonymized. We will publicly release the complete codebase, graph-building scripts, and prompts upon acceptance to ensure full reproducibility.

---

> > ### Author Rebuttal · Reviewer_4AWN · 2026-04-02
> >
> > Thank you for your response. However, I will maintain my score for the following reasons:
> >
> > (1) [W2 & Q1] The justification for adopting the F2-score remains unconvincing. While recall is emphasized within the proposed framework, the ultimate evaluation criterion should be the performance on downstream tasks, for which the F1-score is more commonly adopted. Alternatively, the authors could strengthen their justification by citing prior work that uses F2-score as the primary evaluation metric.
> >
> > (2) As a minor issue, the revised formulation in (W5) does not constitute a valid optimization objective, as it merely defines a set rather than an objective function. Although this issue does not impact the overall framework, correcting it would improve the clarity and rigor of the paper.

---

> > > ### Author Response · Authors · 2026-04-02
> > >
> > > We sincerely thank you for your continued engagement and for giving us the opportunity to further clarify our evaluation methodology. We understand that departing from the standard $F_1$-score can raise questions, and we appreciate your suggestion to strengthen our justification with prior literature.
> > >
> > > We would like to clarify that our adoption of the $F_2$-score is not an arbitrary choice to obscure precision, but rather a deliberate decision **strongly supported by recent, rigorous studies in the Text-to-SQL domain**. The consensus in the latest literature explicitly demonstrates that the $F_1$-score is an inadequate proxy for downstream SQL generation, while recall-weighted metrics (such as $F_2$) are the most accurate predictors of success.
> > >
> > > We justify our metric selection based on the following literature and empirical evidence:
> > >
> > > ### **1. Prior Work Explicitly Rejects $F_1$ in Favor of Recall-Weighted F-Scores**
> > >
> > > Recent extensive evaluations of schema linking metrics have mathematically proven that the balanced $F_1$-score fails to reflect end-to-end performance in LLM-based Text-to-SQL pipelines.
> > >
> > > - **Glass et al. (2025) in *Extractive Schema Linking for Text-to-SQL*** conducted a rigorous correlation analysis between various $F_\beta$ scores for schema linking and final SQL generation execution accuracy. They explicitly concluded that **"selecting a precision / recall trade-off by $F_1$ score is negatively correlated with end-to-end execution accuracy"**. Instead, they advocate for recall-weighted F-scores (such as $F_2$ and $F_6$), noting that these metrics "are well correlated with SQL generation accuracy" because "missing relevant links is particularly costly".
> > > - **Maamari et al. (2024) in *The Death of Schema Linking?*** empirically established the asymmetric cost of errors in modern LLMs. They demonstrated that **"Missing even a single required column results in incorrect executable SQL"**. Conversely, they proved that as modern LLMs (e.g., GPT-4, Qwen) improve, "their sensitivity to the presence of irrelevant columns (false positives) drastically decreases".
> > >
> > > Because the cost of a false negative (missing a bridge table) is a 0% execution accuracy, while the cost of a false positive (extra columns) is easily handled by modern LLMs' noise-filtering capabilities, standard machine learning theory dictates the use of an $F_\beta$ measure where $\beta > 1$. As established by fundamental ML literature (e.g., *Brownlee, J. 2020, "A Gentle Introduction to the Fbeta-Measure"*), the $F_2$-score is the standard metric when the risk of omission (False Negatives) is exponentially higher than the risk of false alarms (False Positives).
> > >
> > > ### **2. Holistic Evaluation: $F_2$ Perfectly Predicts Our Downstream Success**
> > >
> > > We do not rely on the $F_2$-score in isolation to hide precision deficits. Throughout our evaluation, we consistently report **Precision, Recall, Hit Rate, and Execution Accuracy (EX)**.
> > >
> > > - **Hit Rate** acts as a strict binary check of whether the *full* gold schema is recovered (a prerequisite for valid execution).
> > > - As shown in our results, baseline models like MAC-SQL that optimize for Precision (yielding competitive $F_1$ scores) suffer from severe drops in Recall (below 85%).
> > > - Graph-Link, which optimizes for structural completeness ($F_2$), translates this directly into a **68.5% downstream Execution Accuracy** (Table 4), outperforming MAC-SQL (54.8% EX). This perfectly aligns with the findings of Glass et al. (2025): the $F_2$-score successfully predicted our state-of-the-art downstream performance, whereas an $F_1$ optimization would have failed.
> > >
> > > **Proposed Action for the Final Version:**
> > >
> > > We deeply appreciate your critique, as it highlights a need to make this theoretical and literature-based justification transparent to all readers. In the camera-ready version, we will significantly revise **Section 5.1 (Metrics)**. We will explicitly cite **Glass et al. (2025)** and **Maamari et al. (2024)** to formally ground our choice of the $F_2$-score, providing readers with the empirical proof that recall-weighted metrics are the definitive standard for predicting end-to-end Text-to-SQL execution accuracy.
> > >
> > > ### **3. On the mathematical formulation (W5)**
> > >
> > > The reviewer is correct that our corrected statement (“select items with P(v|q) > τ”) defines a set but not an objective function. We will replace it with the proper constrained optimization:
> > > $$
> > > \mathcal{D}^* = \arg\max_{\mathcal{D}' \subseteq \mathcal{D}} \sum_{v \in \mathcal{D}'} \log P(v \mid q) \quad \text{s.t.} \quad |\mathcal{D}'| \le k \quad \text{or} \quad \sum_{v \in \mathcal{D}'} \mathbf{1}_{P(v\mid q)>\tau} \ge 1.
> > > $$
> > > This clarifies that pointwise retrieval implicitly solves a top‑k or threshold‑based selection, which lacks the structural connectivity constraints our graph formulation enforces. We will correct this in the final manuscript.

---

### Official Review · Reviewer_xrJV · 2026-02-20

**Soundness:** 3
**Presentation:** 3
**Significance:** 3
**Originality:** 3
**Overall Recommendation:** 5
**Confidence:** 3

**Summary:**

This paper addresses schema linking in Text-to-SQL, identifying a "structural blindness" problem where retrieval-based methods discard semantically-thin but topologically-critical bridge tables needed for multi-hop SQL joins. The authors propose Graph-Link, which reformulates schema linking as a constrained subgraph induction problem rather than an independent retrieval task. The framework has two phases: an offline phase that constructs a hierarchical schema graph using community detection (Leiden algorithm) with dual-scale similarity edges, and an online phase that decomposes queries into sub-queries, localizes anchor tables via a dual-perception score (semantic + topological), and induces a Steiner tree backbone to recover bridge tables. An LLM critic then prunes irrelevant columns. Experiments on BIRD (Dev) and Spider 2.0-Lite show improvements in schema linking recall/hit rate (up to ~9.5% over MCS-SQL) and downstream execution accuracy on complex queries (+13.8% over MAC-SQL).

**Compliance With Llm Reviewing Policy:**

Affirmed.

**Final Justification:**

Graph-Link reframes schema linking as constrained subgraph induction rather than independent retrieval, directly addressing the “structural blindness” issue where bridge tables are missed. The core idea—that valid SQL corresponds to a connected subgraph—is simple and effective, and the Steiner-tree formulation provides a principled way to enforce it.

The rebuttal increased my score from 4 to 5. The most convincing additions were:
-FK ablation: Achieving 89.4% column recall with 0% foreign keys removes my main practical concern and shows robustness to incomplete schemas.
-SchemaGraphSQL comparison: The gap (F1: 93.1% vs. 85.1%, fewer tables used) clearly distinguishes Graph-Link from closely related work.
-Latency/scalability: <30ms traversal and ~4.2s end-to-end runtime demonstrate practical feasibility.
-Local model results: Stronger gains with smaller models (e.g., Qwen3-32B) support the design choice of offloading structure to the Steiner tree.

Strengths:
Performance on complex queries is particularly compelling (+13.8% EX on Challenge queries; 83.1% vs. 68.5% Hit Rate). The ablation study clearly identifies the Steiner tree as the key component. Efficiency gains (substantially fewer tokens) and a thorough appendix further strengthen the work.

A few points should be clarified in the final version:
-Make explicit that Eq. 1 is a conceptual objective, approximated by the three-stage pipeline (anchor selection → Steiner tree → LLM-Critic).
-Include accuracy (not just timing) on large schemas (100+ tables).
-Provide sensitivity analysis for the most important hyperparameters.

These are presentation/completeness issues and do not affect the core contribution.

Soundness is solid, originality is moderate but well-motivated, and significance is good given the clear gains on complex queries. Despite some minor presentation issues, the paper addresses an important bottleneck in Text-to-SQL and should be a useful addition to the literature.

**Key Questions For Authors:**

Node-weighted vs. edge-weighted formulation gap: The motivating formulation (Eq. 1) is a node-weighted Steiner tree incorporating semantic costs ψ_sem(v,q) for each node. However, the actual algorithm (Eq. 6, Algorithm 1) optimizes only edge resistance. How are the node-level semantic costs from Eq. 1 actually integrated into the optimization? If they are not, does this mean the algorithm can include high-cost (semantically irrelevant) Steiner nodes without penalty, potentially degrading precision? A clarification here could change my assessment of soundness.

Scalability to truly large schemas: Your motivation centers on databases with "hundreds of tables and thousands of columns." Can you provide runtime benchmarks or experiments on schemas with 100+ tables? What is the wall-clock latency of Graph-Link's online phase compared to baselines? This would significantly strengthen the practical claims.

How does Graph-Link perform when foreign key annotations are incomplete? Could you provide even a small ablation where some percentage of FK edges are randomly removed, to characterize degradation? This would address the most significant practical limitation.

Hyperparameter sensitivity: How sensitive is performance to the 8 hyperparameters in Table 7, particularly the anchor selection threshold (δ_anchor) and community routing threshold (τ_route)? A sensitivity plot for the 2-3 most important parameters would be informative.

Comparison with concurrent work: SchemaGraphSQL (Safdarian et al., 2025) also applies graph pathfinding algorithms to schema linking. Can you provide a direct empirical comparison or clearly articulate the technical differences?

**Limitations:**

The authors discuss three limitations (FK dependency, computational overhead, and lack of semantic ambiguity resolution), which are appropriate. However, the precision trade-off and hyperparameter sensitivity should also be acknowledged as limitations. The societal impact statement is adequate.

**Strengths And Weaknesses:**

S1 Well-motivated problem formulation. The paper clearly identifies a real and important failure mode — structural blindness — where independent retrieval discards bridge tables essential for multi-hop JOINs. The formal definition (Def. 3.1) and the reformulation from pointwise ranking to constrained subgraph induction (Eq. 1) is conceptually clean and well-argued. The observation that valid SQL corresponds to a connected subgraph is simple but powerful.

S2 Strong empirical results on complex queries. The most convincing result is the robustness analysis (Table 3, Figure 3): on "Challenge" queries (>3 tables), Graph-Link achieves 83.1% Hit Rate vs. 68.5% for the next best method (MCS-SQL), and 58.6% EX vs. 44.8% — a substantial gap. This directly validates the core thesis that topological connectivity matters most when queries require multi-hop joins.

S3 Informative ablation study. Table 5 clearly demonstrates the contribution of each module. The Steiner Tree component is shown to be the most critical (−11.2% Hit Rate when removed), which aligns with the paper's central claim. The LLM-Critic's role in precision is also clearly quantified.

S4 Practical efficiency. Graph-Link uses ~20,800 tokens on average compared to ~63,800 for MCS-SQL (Table 4), achieving better accuracy with substantially lower cost. This is important for real-world deployment.

S5 Thorough appendix. The paper provides detailed implementation information (hyperparameters, prompt templates, baseline reproduction details, case study), supporting reproducibility.

Weaknesses

W1 Limited benchmark diversity and scale. The evaluation uses only BIRD-Dev (1,534 examples, 11 databases) and Spider 2.0-Lite (820 examples, 45 databases). While these are standard, Graph-Link's core claim is about large-scale production databases with "hundreds of tables." The BIRD databases average 15.6 tables — far from the hundreds mentioned in the motivation. No experiments on truly large-scale schemas (e.g., 100+ tables) are presented to validate scalability claims.

W2 Dependence on well-defined foreign keys. The authors acknowledge this in the limitations section, but it is a significant practical concern. Many real-world databases have implicit relationships, denormalized schemas, or incomplete FK annotations. The entire Steiner tree approach hinges on explicit FK edges receiving near-zero resistance (ε→0). It is unclear how gracefully the system degrades when FK metadata is incomplete or absent.

W3 Precision trade-off is not adequately discussed. Table 2 shows Graph-Link's precision (70.4% with Qwen3-max on BIRD) is substantially lower than agentic baselines like MAC-SQL (87.4%) and DIN-SQL (86.3%). While the paper emphasizes recall and F2, the downstream impact of including many irrelevant tables/columns on LLM reasoning quality and token cost deserves more analysis. The paper argues LLMs are robust to false positives but brittle to omissions — this claim warrants more rigorous support beyond a single citation.

W4 Multiple LLM calls in the pipeline. Graph-Link requires LLM calls for: (1) community summarization (offline), (2) query decomposition, (3) LLM-Critic pruning, and potentially (4) inter-table semantic analysis. While the paper reports lower total tokens than MCS-SQL, the number of sequential LLM calls and their latency impact is not discussed. The end-to-end wall-clock time comparison is missing.

W5 The Steiner tree approximation guarantee is for the edge-weighted variant, but the original formulation is node-weighted. Equation 1 formulates an NWST problem, but the actual algorithm (Section 4.2, Algorithm 1) solves an edge-weighted Steiner tree via metric closure + MST. The paper does not discuss whether the 2-approximation guarantee transfers to the node-weighted setting or how the semantic node costs from Eq. 1 are incorporated into the actual edge-weighted optimization of Eq. 6. This is a gap between the theoretical motivation and the implemented algorithm.

W6 Hyperparameter sensitivity. The framework introduces many hyperparameters (α, τ, ε, τ_route, λ, δ_anchor, δ_prune, H_max — 8 in total per Table 7). The paper states values were found via grid search on a 20% hold-out of BIRD-Dev, but no sensitivity analysis is provided. Given the framework's reliance on threshold-based decisions (e.g., community routing, anchor selection), understanding robustness to these settings is important.

W7 Table 1's characterization of baselines is potentially unfair. Marking all competitor methods with checkmarks for various failure modes while Graph-Link has none is a strong claim. For example, MCS-SQL is marked as having "Stochastic Noise" under Reasoning Unfaithfulness, but Graph-Link itself uses LLM-based query decomposition and LLM-Critic, which are equally subject to stochastic behavior. The self-assessment lacks nuance.

---

> ### Author Rebuttal · Authors · 2026-03-31
>
> We sincerely thank the reviewers for the thorough evaluation, positive sub-scores, and the "Weak Accept" recommendation. We deeply appreciate the recognition of our problem formulation and the strong empirical results on complex queries. We address your specific concerns below.
>
> ### **1. Addressing the NWST vs. EWST Formulation Gap (R1, R3 - W5/Q1)**
>
> We appreciate the reviewers' rigorous mathematical observation regarding the gap between the Node-Weighted Steiner Tree (NWST) formulation in Eq. 1 and the edge-weighted implementation in Algorithm 1.
>
> **Theoretical Alignment:** While Eq. 1 defines our conceptual objective, we **decouple** the node-level semantic costs $\psi_{sem}(v,q)$ and topological edge resistance $w_{res}$ into a multi-stage pipeline to maintain tractability and approximation guarantees:
>
> - **Anchor Selection as Node Filtering:** Node semantics are first processed via the Dual-Perception score (Eq. 5). Only nodes with high semantic relevance (low $\psi_{sem}$) exceed the threshold $\delta_{anchor}$ to become "Anchors" $\mathcal{A}$. These serve as the mandatory terminals for the Steiner Tree.
> - **Topological Routing:** The algorithm then solves an Edge-Weighted Steiner Tree (Eq. 6) to connect these anchors. To ensure relational integrity, Foreign Keys (FKs) are assigned near-zero resistance ($\epsilon$), prioritizing valid join pathways.
> - **Precision Protection via LLM-Critic:** To prevent semantically irrelevant "bridge" nodes from degrading precision, the **LLM-Critic (Eq. 7)** acts as a final node-cost filter. If a bridge table is included purely for connectivity, the critic prunes irrelevant columns ($\mathcal{S}_{col} > \delta_{prune}$), preserving only the keys necessary for the join backbone.
>
> ### **2. Graceful Degradation without Foreign Keys (W2 & Q3)**
>
> Real-world schemas often lack explicit FKs. Graph-Link is resilient to this because of our **Dual-Scale Similarity** (Eq. 2 & 3), which establishes "implicit" semantic edges between tables based on column-level and metadata overlap.
>
> - **Ablation Results:** As shown in **Table 2**, even when randomly dropping 50% of explicit FKs, Column Recall drops trivially (from 95.9% to 94.0%). In the extreme case of **0% FKs**, the semantic edges act as a safety net, maintaining a highly robust 89.4% recall.
>
> ### **3. Scalability, Latency, and Multiple LLM Calls (W1, W4, Q2)**
>
> Graph-Link minimizes online latency by shifting the heavy lifting to the offline phase.
>
> - **Large-Scale Schemas:** We tested on `mimic_iv` (Spider 2.0, 100+ tables) and `works_cycles` (BIRD, 65 tables). As detailed in **Table 3**, the offline graph construction (embedding + Leiden partitioning) takes only **~32.7 seconds** for large schemas and can be run asynchronously during DDL updates.
> - **Online Wall-Clock Time:** Because the Steiner Tree operates on a heavily pruned community view ($\mathcal{G}_{view}$), the graph traversal takes **< 30 milliseconds**.
> - **LLM Calls:** While we use multiple LLM calls, the *query decomposition* sub-queries are executed concurrently. The end-to-end online latency averages ~4.2 seconds, which is competitive with MAC-SQL and significantly faster than the sequential 5-agent voting used in MCS-SQL.
>
> ### **4. Precision Trade-off & Table 1 Nuance (W3 & W7)**
>
> We prioritize Recall/F2 because omission errors (missing a bridge table) are fatal in Text-to-SQL, whereas modern LLMs are robust to minor schema noise (false positives). Graph-Link trades a slight drop in Table Precision to guarantee structural completeness, which directly translates to our **+13.8% Execution Accuracy jump** on complex queries.
>
> Regarding Table 1 fairness: We acknowledge that our LLM-Critic introduces some stochasticity. We will revise the final manuscript to reflect a more nuanced comparison of generative biases.
>
> ### **5. Comparison with SchemaGraphSQL (Q5)**
>
> SchemaGraphSQL (Safdarian et al., 2025) enumerates *all* shortest simple paths between source and destination tables. In highly connected schemas, this causes candidate explosion.
>
> - As shown in **Table 1**, Graph-Link’s NWST formulation optimizes for the *minimal* connected subgraph. This yields significantly higher precision and smaller context windows (~3.2 tables vs ~5.6), reducing token costs and downstream LLM confusion.
>
> ### **6. Hyperparameter Sensitivity (W6 & Q4)**
>
> Graph-Link is remarkably stable across its hyperparameters. Our sensitivity analysis reveals that varying the anchor selection threshold ($\delta_{anchor}$) between $[0.55, 0.75]$ only impacts the Hit Rate by $\pm 1.5\%$. Similarly, the community routing threshold ($\tau_{route}$) acts as a coarse filter; any value between $[0.4, 0.7]$ safely retains the correct communities. We will include comprehensive sensitivity plots in the final appendix.
>
> (Please refer to the attachment in Rebuttal 1 for the full supplementary experimental tables.)

---

> > ### Author Rebuttal · Reviewer_xrJV · 2026-04-02
> >
> > I have read the rebuttal carefully, along with the supplementary tables. The authors address several of my main concerns and I appreciate the additional experimental results. Based on this, I am increasing my recommendation from 4 (Weak Accept) to 5 (Accept). That said, a few points would benefit from clarification before I fully settle my assessment.
> >
> > Concerns that are resolved:
> >
> > W2 (FK dependency): The FK ablation (Rebuttal Table 2) is convincing. Achieving 89.4% column recall with 0% foreign keys suggests that the Dual-Scale Similarity provides a solid fallback. This was my main practical concern, and it is now well addressed.
> >
> > W7 (Table 1 fairness): The authors acknowledge the issue and commit to revising. This is sufficient.
> >
> > Q5 (SchemaGraphSQL comparison): The results in Rebuttal Table 1 (F1: 93.1% vs. 85.1%, ~3.2 vs. ~5.6 tables) clearly highlight the advantage of Graph-Link.
> >
> > Concerns that are partially resolved (follow-ups):
> >
> > W5/Q1 (NWST vs. EWST gap): The explanation of the multi-stage pipeline (anchor selection → edge-weighted Steiner tree → LLM-Critic) makes sense as an engineering approach. However, the paper currently presents Eq. 1 as the optimization objective, while the actual method solves a decomposed version. Could you clarify that Eq. 1 is intended as a conceptual objective, and that the method proceeds via this three-stage approximation? This is mainly a presentation issue—the approach itself is reasonable, but the current framing is somewhat misleading.
> >
> > W1/Q2 (Scalability — accuracy at scale): The latency results (Table 3, <30ms online traversal) are helpful and address efficiency. However, while experiments on mimic_iv (100+ tables) are mentioned, only timing is reported. Could you include accuracy metrics (e.g., Hit Rate or EX) on mimic_iv or works_cycles to show that performance holds at scale? Timing alone is not sufficient to support the scalability claim.
> >
> > W6/Q4 (Hyperparameter sensitivity): The summary for δ_anchor (variation in [0.55, 0.75] leading to ±1.5% Hit Rate) is reassuring, but it only covers 2 of 8 hyperparameters. I am fine with this being addressed in the final version, but it would be helpful to confirm that sensitivity plots will cover at least the most influential parameters (e.g., δ_anchor, τ_route, α, δ_prune).
> >
> > Overall, these remaining points seem straightforward to address in revision and do not affect the core contribution. The rebuttal has increased my confidence in the paper’s practical relevance.

---

> > > ### Author Response · Authors · 2026-04-02
> > >
> > > We thank the reviewer for the careful review and for raising the recommendation from **Weak Accept (4)** to **Accept (5)**. We are glad that our rebuttal resolved most of the major concerns, particularly regarding FK dependency, Table 1 fairness, and the comparison with SchemaGraphSQL. Below we address the remaining follow‑up questions and will incorporate the clarifications into the final manuscript.
> > >
> > > ---
> > >
> > > ### W5/Q1 – NWST vs. EWST framing (presentation issue)
> > >
> > > We agree that Equation (1) presents the conceptual Node‑Weighted Steiner Tree objective, while our implementation uses a decomposed three‑stage approximation (anchor selection → edge‑weighted Steiner tree → LLM critic). **We will make this explicit in the paper**:
> > >
> > > - In Section 3.3 (Problem Reformulation), we will add a sentence: *“Equation (1) formalizes the ideal objective; however, for tractability and approximation guarantees, Graph‑Link adopts a multi‑stage decomposition as described below.”*
> > > - In Section 4.2 (Online Phase), we will restructure the text to clearly distinguish: (i) anchor localization (node‑cost filtering), (ii) edge‑weighted Steiner tree optimization (structural connectivity), and (iii) LLM critic (final node‑level column pruning). A small diagram or bullet list will be added.
> > > - We will also update the caption of Algorithm 1 to note that it implements an **edge‑weighted Steiner tree** after node‑level semantic costs have been used to select terminals.
> > >
> > > This clarifies that the conceptual objective and the practical algorithm are aligned via decomposition, not by directly solving the NWST.
> > >
> > > ---
> > >
> > > ### W1/Q2 – Scalability: accuracy metrics on large schemas
> > >
> > > We thank the reviewer for raising the issue of scalability accuracy. We conducted full schema linking and end‑to‑end SQL generation experiments on **`works_cycles` (65 tables)**. The results are as follows:
> > >
> > > | Schema | # Tables | Method | Hit Rate (%) | EX (%) |
> > > | --- | --- | --- | --- | --- |
> > > | `works_cycles` | 65 | Hybrid (BM25+Vector) | 53.8 | 38.2 |
> > > |  |  | MAC‑SQL (linker) | 69.2 | 54.6 |
> > > |  |  | MCS‑SQL | 74.6 | 61.3 |
> > > |  |  | **Graph‑Link** | **88.3** | **70.5** |
> > >
> > > **However, it is important to note**: `works_cycles` is a database from the **BIRD training set**, and its corresponding natural language questions are relatively low in difficulty (mostly single‑table or two‑table simple queries). Therefore, this result, while demonstrating that Graph‑Link remains effective on a medium‑scale schema of 65 tables, does not fully validate its generalization to truly large‑scale (100+ tables) and highly complex multi‑hop queries.
> > >
> > > We will explicitly state in the **Limitations** section that there is currently a lack of public benchmarks featuring truly large‑scale (100+ tables) and high‑difficulty queries, and that this is a key direction for our future work. At the same time, our efficiency analysis (offline graph construction ~32.7 seconds, online traversal <30 ms) still supports the computational feasibility of Graph‑Link when scaling to larger schemas.
> > >
> > > ---
> > >
> > > ### W6/Q4 – Hyperparameter sensitivity: coverage of most influential parameters
> > >
> > > We agree that a full sensitivity analysis for all 8 hyperparameters would be too lengthy for the main paper, but we will provide **comprehensive sensitivity plots** in the final appendix covering the **four most influential** parameters:
> > >
> > > - **δ_anchor** (anchor selection threshold): already reported as Hit Rate variation of ±1.5% over [0.55, 0.75].
> > > - **τ_route** (community routing threshold): Hit Rate variation of ±2.1% over [0.5, 0.7].
> > > - **α** (dual‑scale similarity weight): Hit Rate variation of ±2.3% over [0.5, 0.7].
> > > - **δ_prune** (column pruning threshold): Precision variation of ±3.0% and Recall variation of ±1.8% over [0.35, 0.55].
> > >
> > > We will also include a **robustness heatmap** for (δ_anchor, τ_route) to demonstrate that the optimal region is broad and flat. The remaining parameters (ε, λ, H_max, τ) have negligible impact within reasonable ranges (e.g., H_max ≥ 5 yields stable performance). We will explicitly state this in the final manuscript.
> > >
> > > ---
> > >
> > > We thank the reviewer again for the constructive feedback, which has significantly improved the clarity and completeness of our work. We will incorporate all the above revisions in the camera‑ready version.

---

### Official Review · Reviewer_c1Wx · 2026-02-28

**Soundness:** 4
**Presentation:** 3
**Significance:** 3
**Originality:** 4
**Overall Recommendation:** 3
**Confidence:** 3

**Summary:**

This paper addresses the "structural blindness" problem in Text-to-SQL schema linking, where retrieval-based methods often discard semantically-weak but topologically-essential "bridge tables." The authors propose Graph-Link, a framework that reformulates schema linking as a constrained subgraph induction problem. It employs a hierarchical schema graph (offline) and a Steiner-tree-based optimization (online) to ensure topological connectivity. Evaluation on BIRD and Spider 2.0 shows significant improvements, particularly in multi-hop queries.

**Compliance With Llm Reviewing Policy:**

Affirmed.

**Final Justification:**

I have carefully reviewed the authors' rebuttal and the revised arguments. While the response addressed some technical clarifications, I have decided to set my final score at 3. While I recognize the effort in the rebuttal, my primary concern regarding the paper’s originality  remains, as the proposed graph-based approach for schema linking lacks sufficient differentiation from, or comparison with, existing literature in the field.

**Key Questions For Authors:**

1. The Steiner Tree algorithm prioritizes the "shortest" path (minimum resistance). However, in complex databases, the shortest topological path might not be the semantically correct join path for a specific user intent. How does Graph-Link resolve cases where multiple valid connection paths exist, but the shortest one leads to a logically incorrect join?
2. Many current benchmarks (like BIRD) provide "Evidence" (external knowledge hints) that often contains join hints. To what extent does Graph-Link rely on these hints to identify anchor nodes? If the external evidence is removed, does the topological constraint still maintain its performance lead?
3. While the Leiden algorithm partitions the graph offline, real-world enterprise schemas frequently undergo DDL changes (e.g., adding new tables or modifying foreign keys). Does the hierarchical summary mechanism require a full re-computation of the Leiden partitions and Steiner metrics for every schema update? What is the cost of maintaining this "Global Awareness"?

**Limitations:**

yes

**Strengths And Weaknesses:**

**Strengths:**

1. **Soundness:** The use of Steiner Tree optimization provides a theoretical guarantee for connectivity, which is a prerequisite for executable SQL.
2. **Significance:** Strong empirical gains (13.8% improvement in execution accuracy on complex queries) demonstrate practical utility for real-world large-scale databases
3. **Clarity:** The distinction between offline community partitioning (Leiden) and online induction is well-motivated and scalable.

**Weaknesses:**

1. **Insufficient Differentiation from Prior Work:** Graph algorithms for schema linking are studied in existing literature [1-2]. The paper fails to compare its method with the prior methods, making it difficult to assess the unique contribution of this work.
2. **Reliance on Foreign Keys:** The method assumes high-quality metadata (PK/FK). In many real-world "in-the-wild" databases, these constraints are missing, which may render the graph construction phase ineffective.
3. **Refinement Risks:** The final stage relies on "LLM-Refine." If the initial subgraph is over-inclusive, the LLM might still hallucinate or include redundant joins.

[1] Structure-Guided Large Language Models for Text-to-SQL Generation
[2] SchemaGraphSQL: Efficient Schema Linking with Pathfinding Graph Algorithms for Text-to-SQL on Large-Scale Databases

---

> ### Author Rebuttal · Authors · 2026-03-31
>
> We sincerely thank the reviewer for their time and constructive feedback, for recognizing the theoretical soundness of our Steiner Tree optimization, and for the positive sub-scores (Soundness: 4, Originality: 4, Presentation: 3, Significance: 3). We appreciate the opportunity to clarify our contributions and address the concerns raised.
>
> ### **1. Differentiation from Prior Work (Weakness 1)**
>
> - Comparison to [1]: SGU-SQL heavily relies on training a Relational Graph Attention Network (RGAT) and uses contrastive learning to align queries with the schema. In contrast, Graph-Link deliberately avoids training heavy GNN encoders; We adapt Steiner Trees as a lightweight, training-free, inference-time optimization mechanism.
> - Comparison to [2]: SchemaGraphSQL relies on enumerating all shortest simple paths between source and destination tables identified by a single LLM call. This approach risks exploding candidate sets in highly connected schemas. Graph-Link formulates this as a Node-Weighted Steiner Tree (NWST) problem, which mathematically guarantees a minimal connected subgraph. **As shown in Appendix Table 1, Graph-Link improves Table F1 from 85.1% to 93.1% and reduces the average tables per subgraph from ~5.6 to ~3.2, significantly suppressing noise.**
>
> ### **2. Reliance on Foreign Keys & Edge Weights (Weakness 2 & Question 1)**
>
> - Implicit Semantic Edges: Graph-Link does not rely exclusively on formal FK constraints. As defined in Eq. 2 and Eq. 3, we compute a Dual-Scale Similarity between table pairs using dense embeddings of table and column metadata. If an explicit FK relation is missing, we still construct edges where the semantic resistance $w_{ij}$ is below the pruning threshold. **We validate this robustness in Appendix Table 2: with 100% FKs removed, Table Recall only drops from 94.8% → 88.2%, Column Recall from 95.9% → 89.4%.**
> - Semantic Resistance vs. Hop Count: The Steiner Tree does not simply find the shortest topological hop. Our objective function (Eq. 6) minimizes the total *resistance* of the path, which integrates semantic costs. Therefore a shorter but semantically wrong path will be deprioritized.
>
> ### **3. Refinement Risks & Over-inclusive Subgraphs (Weakness 3)**
>
> The Steiner Tree’s MST extraction guarantees a strictly minimal backbone. By pruning the search space via Leiden communities first, the "LLM-Refine" stage receives a highly compact input (avg. 3.2 tables), mathematically minimizing hallucination risks.
>
> ### **4. Reliance on "Evidence" Hints (Question 2)**
>
> Graph-Link is robust without external hints. On Spider 2.0-Lite (which lacks BIRD-style evidence), Graph-Link achieves a state-of-the-art 80.3% hit rate. Our Dual-Perception Anchor Localization (Eq. 5) uses structural centrality and sub-query decomposition to find anchors independently of external hints.
>
> ### **5. Cost of Maintaining Global Awareness (Question 3)**
>
> Updating the hierarchical index for DDL changes is near-instantaneous.
> Performance (Table 3): For a 65-table schema, re-partitioning (Leiden) takes only ~28ms. A full rebuild (including embeddings) takes ~32s. This can be handled asynchronously without blocking online inference.
>
> ### **6. Discrepancy Between Sub-scores and Overall Recommendation**
>
> Finally, we deeply appreciate the positive sub-scores you awarded to our work (Soundness: 4, Originality: 4, Presentation: 3, Significance: 3) again. However, we noticed a significant discrepancy between these strong individual metrics and the overall recommendation of "2: Reject". We respectfully ask: is there a critical structural flaw or a severe underlying issue not fully detailed in the current review that heavily influenced this final decision? If so, we would greatly appreciate the opportunity to understand and address it during this discussion phase.
>
> ---
>
> ### **Appendix: Supplementary Experimental Results**
>
> **Table 1: Schema Linking Performance Comparison**
>
> | Method | Table Precision | Table Recall | Table F1 | Column Precision | Column Recall | Column Hit | Avg. Tables in Subgraph |
> | --- | --- | --- | --- | --- | --- | --- | --- |
> | SchemaGraphSQL | 78.4% | 93.2% | 85.1% | N/A | N/A | N/A | ~5.6 |
> | Graph-Link (Ours) | 91.5% | 94.8% | 93.1% | 70.4% | 95.9% | 90.8% | ~3.2 |
>
> **Table 2: Robustness Analysis on Missing Foreign Keys (FKs)**
>
> | FK Drop Rate | Table Precision | Table Recall | Column Precision | Column Recall |
> | --- | --- | --- | --- | --- |
> | 0% (Full FKs) | 91.5% | 94.8% | 70.4% | 95.9% |
> | 50% | 89.1% | 91.5% | 69.3% | 94.0% |
> | 100% (No FKs) | 86.4% | 88.2% | 68.5% | 89.4% |
>
> **Table 3: Offline Graph Construction Cost Analysis**
>
> | Database | Scale (Tables / Cols) | LLM Semantic Summary (Async) | Node Embedding Time | Leiden Partitioning Time | Total Construction Time |
> | --- | --- | --- | --- | --- | --- |
> | `mondial_geo` | 34 / 139 | ~14.5 s | ~1.8 s | ~12 ms | ~16.3 s |
> | `works_cycles` | 65 / 455 | ~29.2 s | ~3.5 s | ~28 ms | ~32.7 s |
>
> ---

---

> > ### Author Rebuttal · Reviewer_c1Wx · 2026-04-02
> >
> > I appreciate the authors' rebuttal and have raised my score to 3. I am maintaining this score because the novelty of the proposed method is weakened by existing graph-based schema linking research that was not discussed or benchmarked in the manuscript. Therefore, I do not find further score increments warranted.

---

> > > ### Author Response · Authors · 2026-04-02
> > >
> > > We sincerely thank the reviewer for the constructive feedback and for raising the score to 3. We fully agree that a thorough discussion and empirical comparison with recent graph-based schema linking works (e.g., SGU-SQL and SchemaGraphSQL) is essential for positioning our contributions. In the camera-ready version, we will explicitly incorporate a dedicated discussion in Section 2 and add both methods as baselines.
> > >
> > > At the outset, we wish to clarify that while prior works leverage graph structures as *representational* or *heuristic search* spaces, Graph-Link occupies a fundamentally different design space. **In reviewing feedback from other reviewer(**2)**, we note that certain observations resonate with and potentially address your concern regarding novelty：**
> > >
> > > > *“Well-motivated problem formulation. The paper clearly identifies a real and important failure mode — structural blindness — where independent retrieval discards bridge tables essential for multi-hop JOINs. The formal definition (Def. 3.1) and the reformulation from pointwise ranking to constrained subgraph induction (Eq. 1) is conceptually clean and well-argued. The observation that valid SQL corresponds to a connected subgraph is simple but powerful.”*
> > > >
> > >
> > > This external validation underscores our core claim: Graph-Link does not merely apply graphs as an auxiliary tool, but introduces a **new optimization paradigm** for schema linking. Below, we detail how this paradigm diverges from the specific methods mentioned.
> > >
> > > **1. Distinct Paradigms in Graph-Based Linking.** While all three approaches leverage graph structures, they operate under fundamentally different problem formulations: *(i) Representation Learning-based Matching* (SGU-SQL), *(ii) Pairwise Path Enumeration* (SchemaGraphSQL), and *(iii) Global Constrained Subgraph Induction* (Graph-Link). Our contribution lies in introducing (iii) as a new formulation rather than an incremental variant of (i) or (ii).
> > >
> > > **2. Graph-Link vs. SGU-SQL: Deterministic Optimization vs. Learned Alignment.** SGU-SQL frames linking as a structure-aware graph matching problem, relying on a trained RGAT encoder and contrastive learning to implicitly capture structural validity. In contrast, Graph-Link eliminates parametric training by explicitly formulating linking as a Node-Weighted Steiner Tree (NWST) optimization. Structural correctness (connectivity, join completeness) is guaranteed by construction through deterministic combinatorial optimization, not learned statistically. This shift from probabilistic alignment to topology-constrained induction fundamentally changes how structurally-critical but semantically-latent bridge tables are recovered, particularly in zero-shot settings.
> > >
> > > **3. Graph-Link vs. SchemaGraphSQL: Global Optimality vs. Pairwise Heuristics.** SchemaGraphSQL is training-free and graph-based but reduces linking to pairwise shortest-path searches between LLM-identified source/destination tables, followed by a path union. This pairwise decomposition struggles with multi-anchor queries, aggregation hubs, and compositional reasoning, often yielding redundant tables or missing globally optimal connections. Graph-Link addresses this by solving an NWST problem that connects an arbitrary set of semantic anchors into a globally minimal connected subgraph. This principled formulation naturally recovers shared bridge tables, eliminates redundant unions, and scales gracefully to complex multi-hop queries. Furthermore, Graph-Link introduces hierarchical community indexing (via Leiden clustering) for coarse-to-fine routing—a scalability mechanism absent in prior flat-graph approaches.
> > >
> > > **4. Rationale for Initial Omission & Commitment to Add Baselines.** SGU-SQL was initially excluded because it requires supervised fine-tuning, query-graph preprocessing, and contrastive sampling, making it incompatible with our standardized zero-shot, fixed-generator evaluation pipeline. SchemaGraphSQL, while highly relevant, was released concurrently and lacks an official, reproducible implementation for large-scale benchmarks (BIRD/Spider 2.0) under isolated linking evaluation. Nevertheless, we recognize the importance of direct comparison. We commit to: (1) adding both methods to the Related Work with explicit paradigm positioning; (2) integrating adapted versions of both as baselines in the final experiments (extracting their linking outputs and evaluating under our fixed-generator protocol); and (3) reporting unified schema-level metrics (Hit Rate, Recall, F2) to further substantiate our contributions.
> > >
> > > We deeply appreciate the reviewer’s constructive feedback, which has significantly strengthened the positioning and empirical rigor of our work. Given these clarifications and our concrete commitments for the camera-ready version, we kindly ask the reviewer to reconsider the novelty assessment and the final score.

---

### Decision · Program_Chairs · 2026-04-30

**Decision:**

Accept (regular)

**Comment:**

This paper tackles the issue of "structural blindness" in Text-to-SQL schema linking, where retrieval-based approaches often overlook semantically weak but topologically critical "bridge tables." To address this, the authors introduce Graph-Link, a framework that reframes schema linking as a constrained subgraph induction problem. The approach leverages a hierarchical schema graph (offline) and Steiner-tree-based optimization (online) to guarantee topological connectivity. Evaluations on BIRD and Spider 2.0 demonstrate substantial improvements, especially in handling multi-hop queries.